# WDR77 inhibits prion-like aggregation of MAVS to limit antiviral innate immune response

Jiaxin Li[1], Rui Zhang[1], Changwan Wang[1,2], Junyan Zhu[1], Miao Ren[1], Yingbo Jiang[1], Xianteng Hou[1], Yangting Du[1], Qing Wu[1], Shishi Qi[1], Lin Li[3], She Chen[3], Hui Yang [4] & Fajian Hou [1,2] ✉

RIG-I-MAVS signaling pathway plays a crucial role in defending against pathogen infection and maintaining immune balance. Upon detecting viral RNA, RIG-I triggers the formation of prion-like aggregates of the adaptor protein MAVS, which then activates the innate antiviral immune response. However, the mechanisms that regulate the aggregation of MAVS are not yet fully understood. Here, we identified WDR77 as a MAVS-associated protein, which negatively regulates MAVS aggregation. WDR77 binds to MAVS proline-rich region through its WD2-WD3-WD4 domain and inhibits the formation of prion-like filament of recombinant MAVS in vitro. In response to virus infection, WDR77 is recruited to MAVS to prevent the formation of its prion-like aggregates and thus downregulate RIG-I-MAVS signaling in cells. WDR77 deficiency significantly potentiates the induction of antiviral genes upon negative-strand RNA virus infections, and myeloid-specific *Wdr77*-deficient mice are more resistant to RNA virus infection. Our findings reveal that WDR77 acts as a negative regulator of the RIG-I-MAVS signaling pathway by inhibiting the prion-like aggregation of MAVS to prevent harmful inflammation.

Innate immune system provides the first line of defense against invading pathogens[1]. To activate innate immunity, host cells must recognize pathogen-associated molecular patterns (PAMPs) using pattern-recognition receptors (PRRs)[2]. Viral nucleic acids, a major class of PAMPs, are detected in host cells upon viral infection. RIG-I-like receptors (RLRs) such as RIG-I sense viral RNA in the cytoplasm and trigger immune signaling through the adaptor protein MAVS (also known as IPS1, VISA, Cardif)[3-7]. MAVS transmits antiviral signaling to TBK1/IKKε and IKKα/β[8], which activate transcriptional factors IRF3/IRF7 and NF-κB respectively, resulting in the expression of type I interferon (IFN-I), proinflammatory cytokines and other effector

genes[9]. IFN-I then activates signaling pathways that induce a group of IFN-stimulated genes (ISGs), leading to an antiviral state in host cells[10].

MAVS plays a critical role as a central hub that connects virus recognition to the innate antiviral immune response[8,11]. Upon sensing viral RNA, RIG-I undergoes conformational changes and oligomerization, exposing its N-terminal 2CARD to bind to the N-terminal CARD domain of MAVS[12,13]. This interaction induces the formation of prion-like aggregates of MAVS, which act as a sensitive trigger for antiviral signaling[11,14-16]. MAVS aggregation is a hallmark of its activation and is essential for its antiviral function[14,17]. However, the detailed molecular mechanisms responsible for MAVS aggregation in cells remain unclear.

[1]State Key Laboratory of Molecular Biology, Shanghai Institute of Biochemistry and Cell Biology, Center for Excellence in Molecular Cell Science, Chinese Academy of Sciences; University of Chinese Academy of Sciences, Shanghai 200031, China. [2]Key Laboratory of Systems Health Science of Zhejiang Province, School of Life Science, Hangzhou Institute for Advanced Study, University of Chinese Academy of Sciences, Hangzhou 310024, China. [3]National Institute of Biological Sciences, Beijing 102206, China. [4]Shanghai Key Laboratory of Brain Function Restoration and Neural Regeneration, Huashan Hospital, Shanghai Medical College, Fudan University, Shanghai 200032, China. ✉e-mail: fhou@sibcb.ac.cn

Activation of the RIG-I-MAVS pathway is crucial for providing protection against pathogen invasion. However, improper activation can lead to chronic cytotoxicity and autoimmune diseases. Inappropriate or persistent MAVS aggregation has been linked to increased production of IFN-I and autoimmunity in a significant fraction of Systemic Lupus Erythematosus (SLE) patients[18]. Therefore, precise regulation of MAVS activation is necessary to avoid potentially harmful immunopathology[19]. During later viral infection, MAVS activity is negatively regulated by various mechanisms. UBXN1/ GPATCH3 competes with TRAF3/TRAF6 to bind to MAVS for signaling attenuation[20,21]. TSPAN6 binds to MAVS in a ubiquitination-dependent manner, interfering with the recruitment of its downstream signaling molecules and inhibiting MAVS activity[22]. Gp78 localized on the endoplasmic reticulum can mediate the proteasomal degradation of MAVS through the contact of mitochondria-endoplasmic reticulum interactions[23]. Moreover, posttranslational modifications (PTMs), including phosphorylation, ubiquitination, O-GlcNAcylation, and arginine methylation, have also been reported to modulate MAVS activity[8,24]. Phospholipase A2 (cPLA2) in astrocyte cytoplasm activates downstream NF-κB molecules through direct interaction with MAVS, promoting the transcription of several cytokines[25]. TRAF3IP3 binds to MAVS and facilitates its recruitment of downstream TRAF3 during viral infection, thereby promoting antiviral response[26]. ER-localized ATP13A1 is required specifically for the stability and activation of MAVS[27].

WDR77, also known as MEP50, is a human WD repeat domain protein and a part of the 20 S methylosome complex that includes PRMT5, which is responsible for the methylation of arginine[28–30]. WDR77 is expressed in various human tissues[31] and can form distinct complexes with other proteins for different functions[31,32]. Additionally, WDR77 acts as a coactivator for both androgen and estrogen receptors and plays a role in hormonal effects during prostate and ovarian tumorigenesis[33,34]. However, it is currently unclear how WDR77 may contribute to innate antiviral immunity and if it is involved in the RIG-I-MAVS pathway.

We conducted an affinity purification experiment to investigate the unknown mechanism of MAVS-mediated innate immunity and found that WDR77 binds to MAVS. Through direct protein-protein interaction, WDR77 represses the prion-like aggregation of MAVS, thereby negatively regulating MAVS activity after viral infection. Deficiency of WDR77 enhances the innate antiviral response to negative-strand RNA virus both in cells and in vivo. Collectively, our study reveals that WDR77 is a critical negative regulator of the antiviral innate immune response.

## Results

### WDR77 is a MAVS-associated protein that dampens IFN-β signaling

To uncover potential MAVS-associated factors that regulate its activity, we generated stable cells expressing Flag-MAVS based on parental HEK293T cells lacking endogenous MAVS, i.e. HEK293T-Flag-MAVS (Fig. 1a) and performed Flag-MAVS pull-down assays using M2 beads (Fig. 1b). Silver staining following SDS-PAGE of the pulled-down proteins revealed two distinct bands with apparent molecular weights of ~70 kDa and 40 kDa (Fig. 1c). Mass spectrometry analysis revealed the presence of MAVS, PRMT5, WDR77 and other proteins, indicating the association of MAVS with the PRMT5-WDR77 complex (Fig. 1d). Co-immunoprecipitation assays confirmed the binding of MAVS with both PRMT5 and WDR77, in addition to the known binding partner TRAF2 (Fig. 1e).

We proceeded to explore the involvement of PRMT5 and WDR77 in MAVS signaling. HEK293T cells were transiently transfected with Flag-PRMT5 and Flag-WDR77 with or without infection by Sendai virus (SeV) (Fig. 1f–h). We found that WDR77, but not PRMT5, significantly reduced the activation of *IFNB1* promoter in response to SeV infection in a dose-dependent manner (Fig. 1f–h). This was further supported by

the observation that Flag-WDR77 also inhibited the activation of *ISRE* and *NFKB1* promoters following SeV infection (Supplementary Fig. 1a–c). Notably, co-expression of Flag-PRMT5 did not enhance the inhibitory effect of Flag-WDR77 on *IFNB1* promoter activation (Fig. 1i and Supplementary Fig. 1d). Furthermore, the expression of WDR77 attenuated *IFNB1* promoter activation induced by various stimuli, such as SeV, vesicular stomatitis virus (VSV), and poly(I:C) (Fig. 1j, k). Collectively, these findings suggested that WDR77 alone, but not PRMT5, has an inhibitory role in RIG-I-MAVS-mediated *IFNB1* induction.

### WDR77 deficiency potentiates antiviral immune response and restricts virus replication

To investigate how WDR77 functions in RIG-I-MAVS signaling, short hairpin RNAs (shRNAs) were designed to target endogenous WDR77 and were introduced into HEK293T cells. The knockdown efficiency of WDR77 was confirmed at both mRNA and protein levels (Supplementary Fig. 2a, b). Notably, shRNAs targeting WDR77 or PRMT5 resulted in concurrent reduction of both proteins, as reported previously due to their mutual dependence on each other[29]. Knockdown of either WDR77 or PRMT5 resulted in a significant increase in *IFNB1* induction upon SeV or VSV infection (Supplementary Fig. 2c). Furthermore, cells stably expressing these shRNAs were obtained, which also showed enhanced *IFNB1* induction in the absence of WDR77 (Supplementary Fig. 2d-g).

Moreover, using the CRISPR/Cas9 technique, we generated three cell lines deficient in WDR77 (*WDR77*[−/−] #1, #2, #3) (Supplementary Fig. 3a). The absence of WDR77 led to a significant increase in the induction of *IFNB1*, as measured by qPCR (Fig. 2a, b) and ELISA (Supplementary Fig. 3b). Additionally, the activation status of TBK1 and IRF3, indicated by their phosphorylation, was significantly increased upon SeV or VSV infection (Fig. 2c). Furthermore, the induction of *IFNB1*, *ISG56*, *CXCL10*, *ISG54*, *CCL5* and *IL6* was upregulated in *WDR77*[−/−] cells stimulated by SeV, VSV, or poly(I:C) (Fig. 2d-f and Supplementary Fig. 3c–e). Consequently, *WDR77*[−/−] cells exhibited enhanced antiviral immunity and were more resistant to VSV proliferation than wild-type (WT) and *MAVS*[−/−] cells (Fig. 2g, h). These findings suggest that WDR77 plays a critical role in modulating RLR-mediated antiviral signaling.

Additionally, we generated *wdr77*-deficient mouse embryonic fibroblast (MEF) cells (Supplementary Fig. 3f) and observed a significant increase in *Ifnb1* induction upon SeV or VSV infections (Supplementary Fig. 3g). We also observed the inhibitory effect of WDR77 on *IFNB1* induction in HeLa cells (Supplementary Fig. 3h, i). To investigate whether WDR77 plays a role in the DNA-sensing pathway such as cGAS-STING signaling, we generated *WDR77*[−/−] HEK293 cell lines (Supplementary Fig. 3j). Consistent with above-mentioned data, we observed a significant upregulation of *IFNB1* induction in *WDR77*[−/−] HEK293 cells in response to SeV, VSV, or poly(I:C) stimulation (Fig. 2i, j). In contrast, the induction of *IFNB1* by Herring testis DNA (HT-DNA) was not affected in the absence of WDR77 (Fig. 2i, j). Furthermore, IRF3 dimer formation, a hallmark of activation of both RIG-I-MAVS and cGAS-STING pathways, was significantly increased in *WDR77*-deficient HEK293 cells, correlating with enhanced *IFNB1* induction (Fig. 2k). Notably, transient expression of wild-type WDR77 suppressed the enhancement of *IFNB1* induction in a dose-dependent manner in *WDR77*[−/−] HEK293T cells (Fig. 2l and Supplementary Fig. 3k). These findings collectively demonstrate that WDR77 negatively regulates IFN-β signaling mediated by the RLRs pathway.

### WDR77 targets MAVS to inhibit IFN-β signaling

In order to investigate the mechanism by which WDR77 suppresses RIG-I-MAVS signaling, an epistasis analysis was performed. HEK293T cells were transiently transfected with WDR77, together with RIG-I(N), MAVS, TBK1, IRF3(S396D), and IKKβ, and the activation of *IFNB1* was measured. The results showed that overexpression of

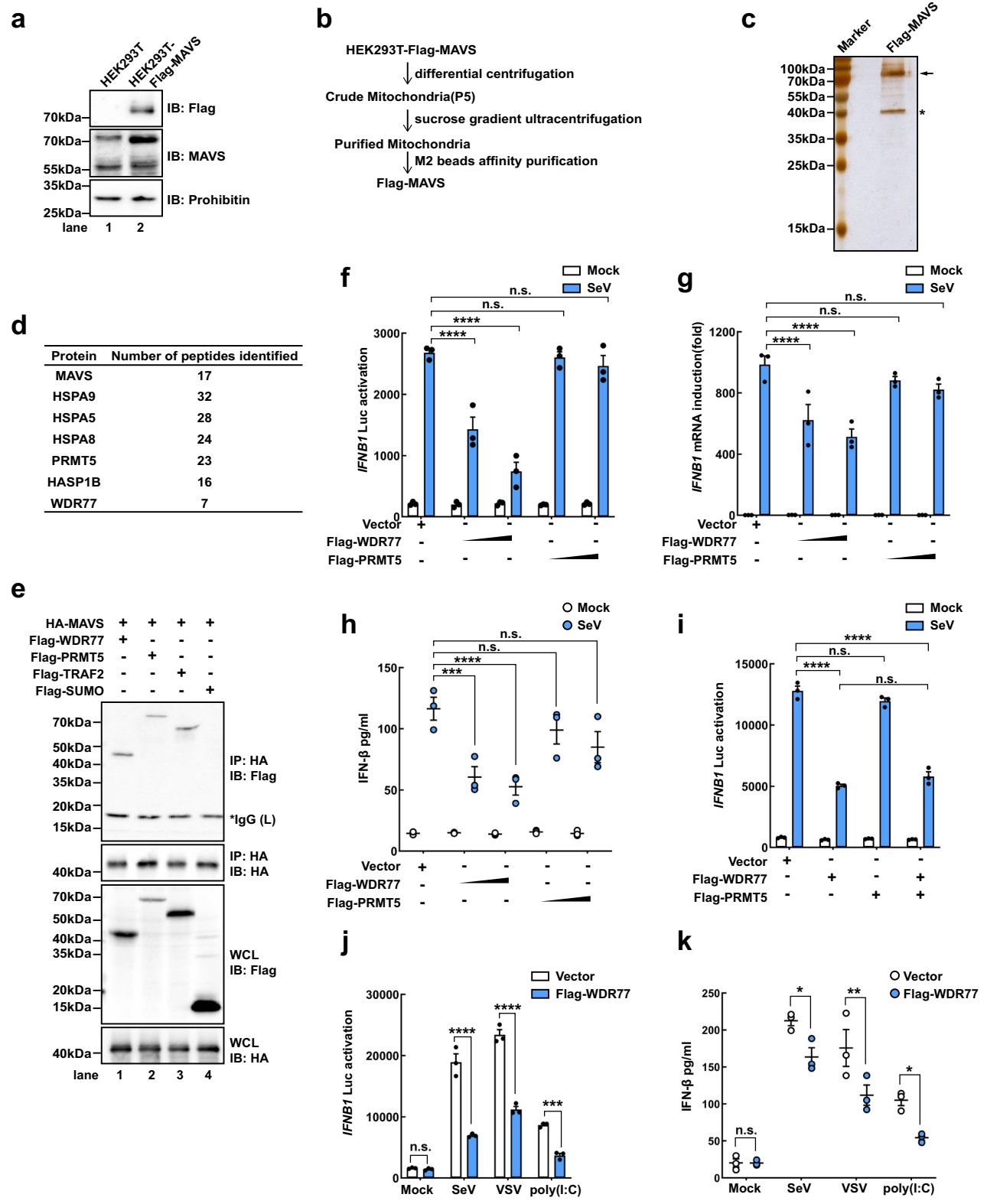

WDR77 crippled *IFNB1* activation by RIG-I(N) and MAVS, but had no effect on *IFNB1* activation by TBK1, IRF3(S396D), or IKKβ (Fig. 3a–c and Supplementary Fig. 4a). Furthermore, WDR77 depletion potentiated the activation of *ISRE* reporter induced by RIG-I(N) or MAVS, but had no effect on *ISRE* activation induced by TBK1 or IRF3(S396D) (Supplementary Fig. 4b, c). These findings suggest that WDR77 functions upstream of TBK1, likely at or downstream of MAVS, in the RIG-I pathway. To determine whether WDR77 inhibits *IFNB1* induction

through MAVS, *WDR77* and *MAVS* double knockout cells (DKO) were generated (Supplementary Fig. 4d). The results demonstrated that the enhancement of interferon induction in the absence of WDR77 is indeed dependent on MAVS, as there was a complete loss of *IFNB1* induction upon SeV stimulation in both *MAVS*[−/−] and DKO cells (Supplementary Fig. 4e, f).

Consistent with the aforementioned results showing that MAVS binds to WDR77, co-immunoprecipitation experiments demonstrated

**Fig. 1 | WDR77 binds to MAVS and negatively regulates IFN-β induction.**
**a** Immunoblot analysis of MAVS expression in P5 fraction containing crude mito-chondria. **b** Purification procedure for Flag-MAVS. **c** Silver staining of purified Flag-MAVS. The Flag-MAVS band is indicated by an arrow and the asterisk indicated interested bands. **d** Peptides identified by mass spectrometry. **e** Plasmids as indi-cated were co-transfected into HEK293T cells. 36 h after transfection, cells were harvested for immunoprecipitation and immunoblotting. Asterisk indicated non-specific bands. **f–h** HEK293T cells were transfected with luciferase reporters and increasing amounts of plasmids as indicated for 24 h, and then stimulated with or without SeV for 12 h. Cells were collected and *IFNB1* promotor activation was detected by luciferase assay (**f**). *IFNB1* induction was measured by quantitative PCR (qPCR) (**g**). Culture medium was collected and IFN-β was detected by ELISA (**h**) (For **f**, *IFNB1*: ****$p < 0.0001$, ****$p < 0.0001$, ns$p = 0.9540$, ns$p = 0.4002$ in sequence; For **g**, *IFNB1*: ****$p < 0.0001$, ****$p < 0.0001$, ns$p = 0.6511$, ns$p = 0.1233$ in sequence; For

**h**, *IFNB1*: ***$p = 0.0002$, ****$p < 0.0001$, ns$p = 0.6413$, ns$p = 0.0503$ in sequence). **i** HEK293T cells were transfected with luciferase reporters and plasmids as indi-cated for 24 h, and then stimulated with or without SeV for 12 h. *IFNB1* promotor activation was analyzed by luciferase assay (For **i**, *IFNB1*: ****$p < 0.0001$, ns$p = 0.0861$, ****$p < 0.0001$, ns$p = 0.1511$ in sequence). **j–k** HEK293T cells were transfected with luciferase reporters and Flag-WDR77 plasmids for 24 h, and then stimulated with or without SeV, VSV or poly(I:C) for 12 h. *IFNB1* promotor activation was analyzed by luciferase assay (**j**). IFN-β was detected by ELISA in (**k**) (For **j**, *IFNB1*: ns$p = 0.9994$; ****$p < 0.0001$, ****$p < 0.0001$, ***$p = 0.0001$ in sequence; For **k**, IFN-β: ns$p > 0.9999$, *$p = 0.0377$, **$p = 0.0059$, *$p = 0.0312$ in sequence). Data are representative of one experiment (**c** and **d**), two independent experiments with similar results (**e**), or three independent experiments (**f–k**). *P*-values were determined by two-way ANOVA (Dunnett's test) (**f**) or two-way ANOVA (Šídák's test) (**g**, **h**, **i**, **j** and **k**). Source data are provided as a Source Data file.

that WDR77 interacts specifically with MAVS but not with RIG-I, TBK1, or IRF3 (Fig. 3d). Interestingly, WDR77 also co-immunoprecipitated with TRAF3, but not TRAF2, TRAF5, or TRAF6 (Fig. 3d). Furthermore, immunofluorescence analysis showed that WDR77 colocalizes with MAVS in cells, and the interaction between WDR77 and MAVS was enhanced upon virus infection (Fig. 3e). These findings suggest that WDR77 may act on MAVS to negatively regulate IFN-β induction.

MAVS is composed of three domains: an N-terminal CARD domain, a middle proline-rich region, and a C-terminal transmembrane domain (amino acids 1-90, 91-172, and 513-540, respectively)[6,35] (Fig. 3f, top). Deletion of the MAVS proline-rich region ablated its interaction with WDR77, suggesting that this region is necessary for the interac-tion (Fig. 3f, bottom). To determine which part of WDR77 binds to MAVS, we created WDR77 mutants lacking various WD40 domains (Fig. 3g, top). We found that WD2-WD3-WD4 domain is essential for its binding to MAVS (Fig. 3g, bottom). These findings suggest that the direct interaction between WDR77 and the proline-rich region of MAVS is mediated by its WD2-WD3-WD4 domain. Strikingly, WDR77 dele-tions containing the WD1-WD2-WD3-WD4 domain, such as Flag-WDR77(1-207) and Flag-WDR77(1-249), could inhibit *IFNB1* production induced by SeV to the same extent as full-length WDR77 (Fig. 3h and Supplementary Fig. 4g, h). We also noticed that Flag-WDR77(70-342) without WD1 domain did not show inhibition on *IFNB1* induction. In summary, our data suggest that WDR77 inhibits IFN-β induction by directly interacting with MAVS through its WD2-WD3-WD4 domain.

## WDR77 inhibits MAVS aggregation and activation
We then went to examine how the interaction between MAVS and WDR77 is modulated during viral infection. In the absence of SeV infection, Flag-WDR77 bound to HA-MAVS, and upon SeV infection, the association between Flag-WDR77 and HA-MAVS was substantially increased (Fig. 4a). Furthermore, enhanced interaction was observed between endogenous WDR77 and MAVS upon stimulation (Fig. 4b, c). On the other hand, the interaction between WDR77 and TRAF3 was not influenced by virus infection (Supplementary Fig. 5a, b). Additionally, WDR77 was not required for TRAF3 recruitment to MAVS, suggesting that TRAF3 might not be involved in the interaction between WDR77 and MAVS (Fig. 4b). Furthermore, we examined the interaction between MAVS and WDR77 at different time points post virus infec-tion. Our findings indicated that the MAVS-WDR77 interaction was augmented at the early stage from zero to 16 h, and then declined at the later stage from 16 to 32 h (Fig. 4c). Notably, the association between WDR77 and RIG-I or MDA5 was not detected throughout the entire time course examined. Based on these observations, we spec-ulate that WDR77 may specifically participate in dampening MAVS activity once an effective innate immune response has been reached to prevent deleterious inflammation.

MAVS is known to form prion-like aggregates to mediate anti-viral signaling[14]. Overexpression of HA-MAVS leads to its activation and prion-like aggregation independent of stimulation, which can be

detected through semi-denaturing detergent agarose-gel electro-phoresis (SDD-AGE) (Fig. 4d and Supplementary Fig. 5c, d). Our results showed that HA-MAVS aggregation was suppressed by Flag-WDR77 in a dose-dependent manner. Similarly, prion-like aggrega-tion of endogenous MAVS induced by SeV infection was dampened by WDR77 expression (Fig. 4e and Supplementary Fig. 5e). As a result, MAVS downstream signalings, including IRF3 phosphoryla-tion and dimerization, were all attenuated (Fig. 4f and Supplemen-tary Fig. 5f). Furthermore, MEF cells deficient in *Wdr77* displayed increased MAVS aggregation upon viral infection, indicating a con-served inhibitory mechanism of WDR77 in antiviral signaling from mouse to human (Fig. 4g). These findings suggest that WDR77 acts as a negative regulator of RLR signaling by repressing MAVS aggregation.

We next investigated the role of WDR77 in the regulation of MAVS aggregation over time, and we performed a time course analysis of MAVS aggregation following virus infection. In wild type cells, MAVS aggregation increased from zero to 16 h and then decreased (Fig. 4h). Notably, in WDR77-deficient cells, MAVS aggregation displayed a similar pattern, but decreased at a slower rate after 16 h. We also observed a correlation between *IFNB1* induction and MAVS aggrega-tion in all samples examined (Fig. 4i and Supplementary Fig. 5g). These findings provide further support for our earlier hypothesis that WDR77 plays a role in downregulating MAVS activity following an effective innate immune response.

## WDR77 impedes MAVS to form prion-like filaments
The aggregation and activation of MAVS in cells have been reported to depend on its ubiquitination[36]. However, our data showed that over-expression of WDR77 did not have an effect on MAVS ubiquitination (Supplementary Fig. 6a). This suggests that WDR77 inhibits MAVS aggregation through a mechanism that is independent of its ubiquiti-nation. To further understand how WDR77 may inhibit MAVS aggre-gation, we performed an in vitro reconstitution assay to examine its inhibitory effect on MAVS activation. HEK293T cells were separated into to S5 and P5 fractions as previously reported[37], where the S5 fraction contains cytoplasmic components such as RIG-I, TBK1, IRF3 and WDR77, while the P5 fraction contains organelles like mitochon-dria, where MAVS resides (Supplementary Fig. 6b). The assay was then performed in two steps: in step 1, S5 fraction from SeV-infected cells was incubated with P5 fraction from uninfected cells, and in step 2, P5 fraction isolated from step 1 was further incubated with S5 from uninfected cells to examine MAVS aggregation and IRF3 dimerization (Supplementary Fig. 6c). P5 fractions from wild type or WDR77-deficient cells showed no difference in MAVS activation (Fig. 5a). However, the S5 fraction from *WDR77*−/− cells infected with SeV was more effective in inducing MAVS aggregation in P5 fractions (Fig. 5b), resulting in more pronounced IRF3 dimerization. These results suggest that WDR77 in the S5 fraction has an inhibitory role in triggering the formation of MAVS aggregates.

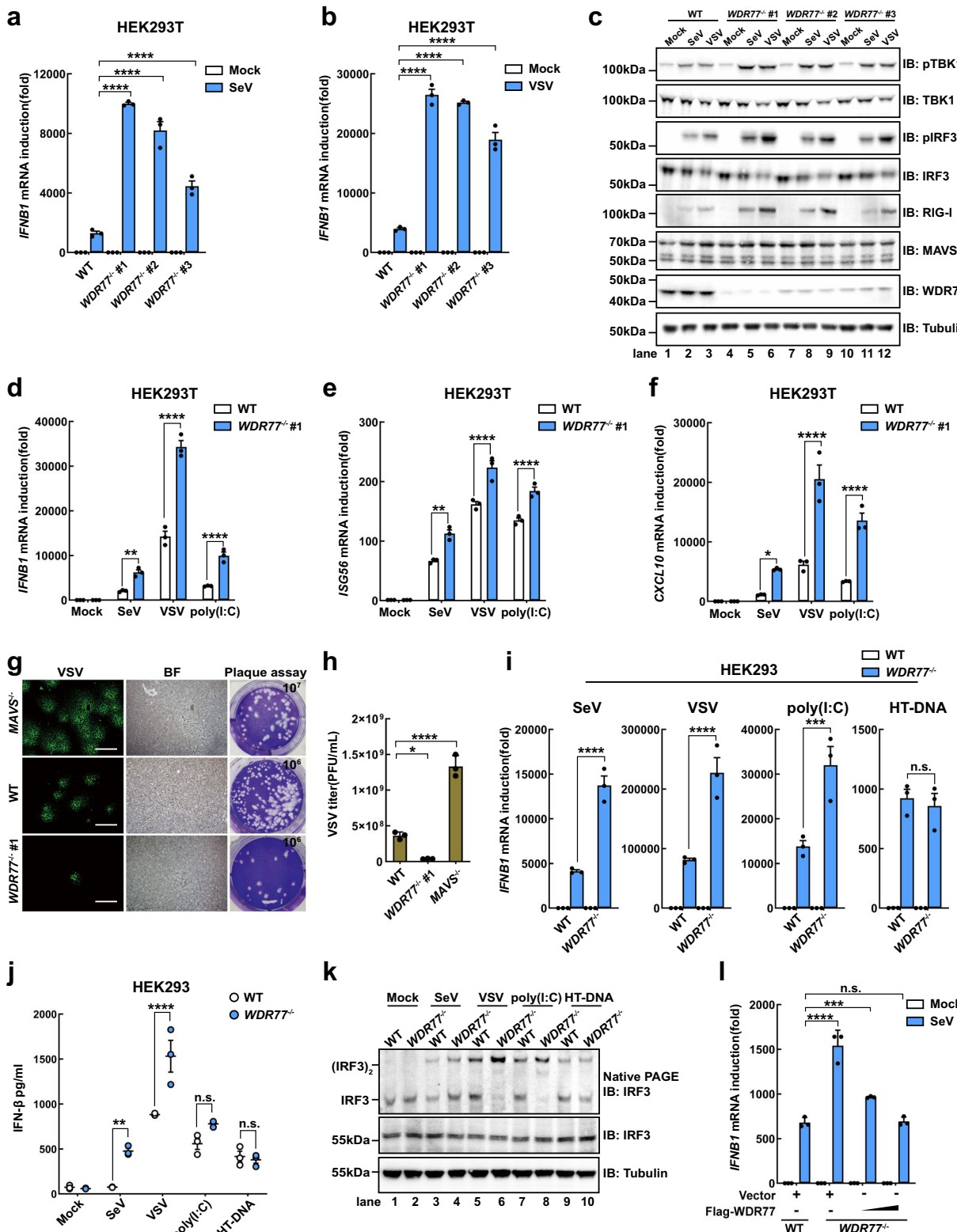

Furthermore, we purified recombinant MAVS as described previously, except that the proline-rich region is included in this study (Supplementary Fig. 6d)[14]. The recombinant MAVS was separated into two fractions, Peak-I and Peak-II, by size-exclusion chromatography (Supplementary Fig. 6f). According to the previous report, Peak-I contained high molecular weight MAVS filaments, while Peak-II consisted of low molecular weight MAVS particles. MAVS in Peak-II can form prion-like filament spontaneously over time[14]. We next incubated

each MAVS fraction with recombinant MBP-WDR77 (Supplementary Fig. 6e) and observed the filament formation under electron microscopy. Consistent with previous reports, the spontaneous aggregation of MAVS in Peak-II fraction was observed after incubation for over 24 h (Fig. 5c)[14,38]. However, recombinant MBP-WDR77 protein effectively prevented the spontaneous aggregation of MAVS in Peak-II fraction. Furthermore, the incubation of MBP-WDR77 with Peak-I promoted disassembly of MAVS filament (Fig. 5d). These results suggested that

**Fig. 2 | WDR77 deficiency potentiates antiviral responses and restricts virus replication. a, b** WT or *WDR77*⁻/⁻ HEK293T cells were stimulated with SeV (**a**) or VSV (**b**) for 12 h before *IFNB1* induction was measured by qPCR (For **a** and **b**, *IFNB1*: all ****$p < 0.0001$). **c** WT or *WDR77*⁻/⁻ HEK293T cells were stimulated with SeV or VSV for 12 h. Cells were harvested and subjected to immunoblotting. **d–f** WT or *WDR77*⁻/⁻ #1 HEK293T cells were stimulated with SeV, VSV or poly(I:C) for 12 h before *IFNB1* (**d**), *ISG56* (**e**) and *CXCL10* (**f**) induction was measured by qPCR (For **d**, *IFNB1*: **$p = 0.0046$, ****$p < 0.0001$, ****$p < 0.0001$ in sequence; For **e**, *ISG56*: ***$p = 0.0001$, ****$p < 0.0001$, ***$p = 0.0001$ in sequence; For **f**, *CXCL10*: *$p = 0.0261$, ****$p < 0.0001$, ****$p < 0.0001$ in sequence). **g** WT, *MAVS*⁻/⁻ or *WDR77*⁻/⁻ #1 HEK293T cells were infected with VSV-GFP (MOI = 0.05) for 12 h before microscopic imaging (left panel) and plaque assay (right panel) were performed. The dilution fold of VSV is displayed in the upper right corner of the plaque analysis. Scale bars indicate 100 μm. **h** GFP-VSV titers were quantitated in plaque assay (For **h**, VSV: *$p = 0.0123$, ****$p < 0.0001$ in sequence). **i–k** WT or *WDR77*⁻/⁻ HEK293 cells were stimulated with SeV, VSV, poly(I:C) or HT-DNA for 12 h before *IFNB1* induction was measured by qPCR (**i**). Culture medium was collected and IFN-β was detected by ELISA (**j**). Following subcellular fractionation, S5 fractions were subjected to native PAGE to examine IRF3 dimerization (**k**) (For **i**, *IFNB1*: ****$p < 0.0001$, ****$p < 0.0001$, ***$p = 0.0007$, ns $p = 0.7439$ in sequence; For **j**, IFN-β: **$p = 0.0014$, ****$p < 0.0001$, ns $p = 0.1199$, ns $p = 0.9973$ in sequence). **l** WT or *WDR77*⁻/⁻ #1 HEK293T cells were transfected with plasmids as indicated for 24 h, and then stimulated with or without SeV for 12 h. *IFNB1* induction was measured by qPCR (For **l**, *IFNB1*: ****$p < 0.0001$, ***$p = 0.0004$, ns $p = 0.9957$ in sequence). Data are representative of three independent experiments with similar results (**c**, **g** and **k**) or three independent experiments (**a**, **b**, **d–f**, **h–j** and **l**). *P*-values were determined by two-way ANOVA (Šídák's test) (**a**, **b**, **d–f**, **i**, **j**), ordinary one-way ANOVA (Tukey's test) (**h**) or two-way ANOVA (Tukey's test) (**l**). n.s. indicates no statistical significance. Source data are provided as a Source Data file.

WDR77 prevents MAVS filament formation through direct protein-protein interaction.

## WDR77 deficiency enhances antiviral immune response in mouse primary cells

To investigate the regulatory function of WDR77 in primary cells, we generated *Wdr77*^fl/fl^ mice by CRISPR-Cas9-mediated gene editing, where two loxP sites flank exons 5 and 9 of the *Wdr77* gene (Supplementary Fig. 7a). WDR77 is highly expressed in the spleen and thymus, indicating a potential immune-related function (Supplementary Fig. 7b). We then crossed these mice with Lyz2-Cre transgenic mice to create myeloid-specific *Wdr77* knockout mice. Successful targeting of *Wdr77* was confirmed by sequencing the PCR fragments (Supplementary Fig. 7c) and immunoblotting. We analyzed WDR77 expression in peritoneal macrophages (PEM) and bone marrow-derived macrophages (BMDM) from *Wdr77*^fl/fl^ Lyz2-Cre+ (referred to as *Wdr77*^CKO^) and WT mice (Supplementary Fig. 7d–f).

We collected PEMs from WT, *Wdr77*^CKO^ and *Mavs*⁻/⁻ mice respectively. PEMs were then subjected to various treatments as depicted. As previously mentioned, we observed an increase in IFN-β secretion in *Wdr77*^CKO^ PEMs following SeV, VSV, and poly(I:C) treatments (Fig. 6a). The expression levels of *Ifnb1*, *Ifna4*, *Isg56*, *Cxcl10* and *Il6* were all elevated in *Wdr77*^CKO^ PEMs, while *Ifnb1* induction was mostly undetectable in *Mavs*⁻/⁻ PEMs (Fig. 6b–f). However, the IFN-β secretion and *Ifnb1* expression induced by HSV-1 or HT-DNA were not influenced in *Wdr77*^CKO^ macrophages, indicating the specific role of WDR77 in RLR signaling (Fig. 6a–f). We also isolated bone marrow-derived macrophages (BMDMs) from mice with different genotypes and conducted similar experiments. Upon treatment with SeV, VSV, and poly(I:C), IFN-β secretion and induction of *Ifnb1*, *Ifna4*, *Isg56*, *Cxcl10* and *Il6* were significantly increased in *Wdr77*^CKO^ BMDMs compared with the WT control (Fig. 6g and Supplementary Fig. 7g–k). Additionally, MAVS aggregation, IRF3 dimerization, and phosphorylation were enhanced in *Wdr77*^CKO^ PEMs upon virus infection (Fig. 6h). In summary, these findings suggest that WDR77 specifically regulates IFN-β signaling mediated by RLR pathways.

### *Wdr77* regulates antiviral innate immunity in mice

To examine the role of WDR77 in antiviral immunity in vivo, we injected WT and *Wdr77*^CKO^ mice with VSV via the tail vein. Upon VSV infection, induction of *Ifnb1*, *Ifna4* and ISGs in various organs (spleen, liver and lung) was significantly higher in *Wdr77*^CKO^ mice compared with WT mice (Fig. 7a, b, and Supplementary Fig. 8a–c). Additionally, *Wdr77*^CKO^ mice showed significantly reduced proliferation of VSV (Fig. 7c) and higher levels of serum IFN-β in response to VSV infection (Fig. 7d). Furthermore, hematoxylin and eosin (H&E) staining revealed more severe inflammation and immune cell infiltration in the lungs of WT mice compared with *Wdr77*^CKO^ mice after VSV infection (Fig. 7e). Consistently, *Wdr77*^CKO^ mice were less susceptible to VSV infection than WT mice (Fig. 7f).

To investigate the function of WDR77 in immune signaling further, we infected mice with influenza A virus (IAV). *Wdr77*^CKO^ mice showed lower mortality than WT mice following IAV infection (Fig. 7g). In contrast, there was no difference in survival between *Wdr77*^CKO^ and WT mice following HSV-1 infection (Fig. 7i). Moreover, the production of IFN-β and induction of *Ifnb1*, *Ifna4* in response to HSV infection were not affected by the absence of *Wdr77*, confirming that WDR77 might not be involved in the DNA-sensing pathway (Fig. 7h and Supplementary Fig. 8d–f). Collectively, these data suggest that WDR77 is an essential negative regulator of the antiviral immune response against negative-strand RNA viruses in vivo.

## Discussion

To maintain proper immune homeostasis, antiviral signaling must be precisely regulated. In RLR pathway, a series of crucial proteins are activated in response to viral infection and must be suppressed by distinct mechanisms to terminate the immune response once the pathogen is eliminated. MAVS is a critical adapter protein in the RLR pathway that forms prion-like aggregates, linking viral RNA sensing to downstream signaling activation during virus infection[39]. Our study demonstrates that WDR77, a member of the WD40 family, functions as an important negative regulator of innate antiviral immunity. WDR77 inhibits the prion-like aggregation of MAVS by binding to its proline-rich region, thereby decreasing RLR-mediated IFN-β production and preventing harmful inflammation (Supplementary Fig. 8g).

In our previous study, we discovered a mechanism that prevents spontaneous aggregation of MAVS in cells[38]. However, how the aggregation of MAVS is regulated after an effective antiviral response remains unclear. To investigate this, we used an affinity purification method to search for unknown factors that may associate with MAVS and regulate its activity. We identified the arginine methyltransferase PRMT5 and its cofactor WDR77 as factors that bind to MAVS. It has been reported that PRMT7 and PRMT9, but not PRMT5, covalently modify MAVS to negatively regulate its activity[40,41]. In agreement with these reports, we found that overexpression of PRMT5 had no effect on IFN-β induction mediated by RLR signaling. Surprisingly, overexpression of WDR77 significantly inhibited IFN-β induction in response to SeV infection. Moreover, we observed that WDR77 deficiency resulted in enhanced activation of TBK1/IRF3 and upregulation of IFN-β induction in HEK293T, MEF cells, and HeLa cells upon negative-strand RNA virus but not DNA stimulation. These findings suggest that WDR77 is an important negative regulator specifically involved in the RIG-I-MAVS pathway.

WDR77 is a WD40 repeat-containing protein with gly-his and trp-asp tandem repeats (GH-WD) in its structure[42–44]. It interacts with MAVS through its WD2-WD3-WD4 domain, and this interaction requires the proline-rich region of MAVS. Intriguingly, while both WDR77 fragments (1-207) and (70-342) encompass the WD2-WD3-WD4 domain, only the (1-207) fragment inhibits IFN-I induction in cells. One plausible explanation is that WDR77(1-207) includes the WD1

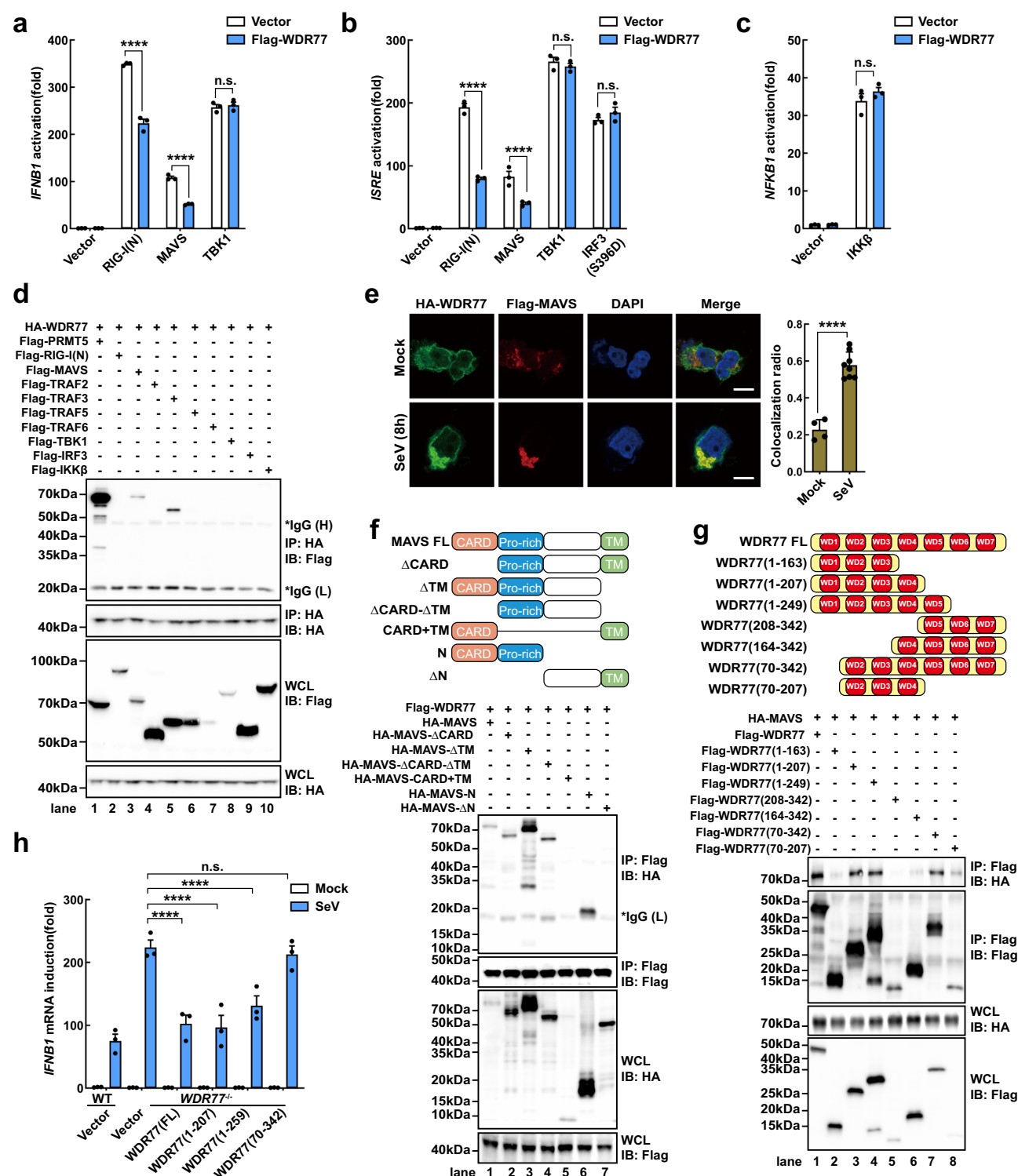

domain, which potentially contributes to the binding of WDR77 to MAVS, surpassing a threshold necessary for WDR77 to inhibit MAVS aggregation. An alternative possibility is that the WD1 domain of WDR77 may have a role in its inhibitory function, in addition to its binding to MAVS. Additionally, our research demonstrates that recombinant WDR77 can effectively impede the prion-like aggregation of MAVS in vitro, highlighting a mechanism facilitated by direct protein-protein interaction. Furthermore, our study showed that WDR77 is recruited to MAVS upon virus infection and prevents the formation of MAVS prion-like filaments, indicating its critical role in regulating IFN-β induction. Notably, mice lacking *WDR77* in myeloid

cells were less susceptible to negative-strand RNA virus infection and exhibited enhanced transcription of IFN-I and other antiviral genes. These findings suggest that WDR77 serves as a crucial negative regulator of innate antiviral immunity. In contrast, another WD repeat-containing protein, WDR5, was shown to promote MAVS-mediated antiviral signaling[45].

WDR77 has been implicated in various cancers, such as lung[46], brain[47], lymph[48], prostate[49], ovarian[34] and breast[50] cancers, and is associated with enhanced tumor growth and poor disease prognosis. Our study suggests that dysfunction of WDR77 could lead to inflammation, which may contribute to the development of cancer since

**Fig. 3 | WDR77 interacts with MAVS and inhibits its activity. a–c** HEK293T cells were transfected with luciferase reporters and plasmids as indicated for 24 h. Cells were collected and *IFNB1* (**a**), *ISRE* (**b**) and *NFKB1* (**c**) promotor activation were detected by luciferase assay (For **a**, *IFNB1*: ****$p < 0.0001$; ****$p < 0.0001$; ns$p = 0.9496$ in sequence; For **b**, *ISRE*: ****$p < 0.0001$; ****$p < 0.0001$; ns$p = 0.8583$, ns$p = 0.4715$ in sequence; For **c**, *NF-KB1*: ns$p = 0.2570$ in sequence). **d** HEK293T cells were transfected with HA-WDR77, Flag-PRMT5, RIG-I, MAVS, TRAF2, TRAF3, TRAF5, TRAF6, TBK1, IRF3 or IKKβ-expressing plasmids as indicated for 36 h. Cells were collected and subjected to immunoprecipitation assay and immunoblotting. Asterisk indicated nonspecific bands. **e** Immunofluorescence assay of co-localization of HA-WDR77 (green) with Flag-MAVS (red) in HEK293T cells following SeV infection for 8 h. Nuclei were stained with DAPI (blue). Scale bars indicate 10 μm (Colocalization radio: ****$p < 0.0001$). **f** Diagram of MAVS truncations used in Co-IP experiments (top panel). Various MAVS truncations were co-expressed with Flag-WDR77 in HEK293T cells for 36 h before immunoprecipitation and immunoblotting (bottom panel) were performed. Asterisk indicates nonspecific bands. **g** Diagram of WDR77 truncations used in Co-IP experiments (top panel). Various WDR77 truncations were co-expressed with HA-MAVS in HEK293T cells for 36 h before immunoprecipitation and immunoblotting (bottom panel) were performed. **h** WT or *WDR77*[-/-] #1 HEK293T were transfected with various WDR77 truncations for 24 h, and then stimulated with or without SeV for 12 h. *IFNB1* induction was measured by qPCR (*IFNB1*: all ****$p < 0.0001$; ns$p = 0.9998$). Data are representative of three independent experiments with similar results (**d**, **e** (left), **f** and **g**), or three independent experiments (**a–c**, **e** (right) and **h**) (mean ± SD of three biological replicates). *P*-values were determined by two-way ANOVA (Šídák's test) (**a–c**, **h**) or unpaired two-sided *t*-test (**e**). n.s. indicates no statistical significance. Source data are provided as a Source Data file.

chronic inflammation is known to be a potential cause of tumor growth. In summary, our results not only shed light on the mechanism by which WDR77 inhibits prion-like aggregation of MAVS to suppress IFN-β induction in antiviral immune response but also provide insights into the clinical relevance of WDR77 in cancer biology.

Firstly, it is important to acknowledge the limitations of this study. While we have demonstrated the interaction between WDR77 and MAVS, along with its inhibitory effect on MAVS aggregation, further investigations using structural analysis would provide valuable mechanistic insights into the specific interaction between WDR77 and MAVS, elucidating how WDR77 effectively hinders MAVS aggregation. Secondly, it is worth noting that this study does not provide evidence linking WDR77 with other regulators that may confer posttranslational modifications on MAVS. Exploring potential connections and interactions between WDR77 and such regulators would be an intriguing avenue for future research. Additionally, the potential involvement of dysregulated WDR77 in virus infection-related diseases remains to be determined. Future investigations are needed to assess whether any associations exist and to shed light on the implications of WDR77 dysregulation in such conditions. These areas warrant further exploration in subsequent studies.

## Methods

### Ethics Statement

All animal experiments were performed in accordance with the guidelines of the Institutional Animal Care and Use Committee (IACUC) at the Shanghai Institute of Biochemistry and Cell Biology, CAS (Approved No. S346-1704-004). The mice were maintained in a specific pathogen-free (SPF) environment, with appropriate temperature and humidity settings (kept at 25°C and suitable humidity, typically around 50%). They were also subjected to a 12-h dark/light cycle and fed with sufficient water and food.

### Mice

*Wdr77*[fl/fl] mice in C57BL/6 background were created by flanking the fifth to ninth exons of *Wdr77* with two loxP sites through CRISPR-Cas9-mediated gene editing (Cyagen Biosciences). *Lyz2-Cre* transgenic mice expressing Cre recombinase were generously provided by Dr. Hong-yan Wang (SIBCB). *Mavs* knockout mice were generated as described[26]. *Wdr77*[fl/fl] mice were crossed with *Lyz2-Cre* mice to get F1 generation. *Wdr77*[fl/fl] *Lyz2-Cre +* (*Wdr77*[CKO]) and *Wdr77*[fl/fl] *Lyz2-Cre-* (control) littermates from the F2 generation were used as future breeders. The genotyping was performed by standard PCR.

### Plasmids and reagents

Human complementary DNA was synthesized using total RNA extracted from HEK293T cells as a template. cDNAs of WDR77, PRMT5, MAVS, RIG-I(N), TRAF3, IRF3 S396D, TBK1 and IKKβ were subcloned into pcDNA3-FLAG or pcDNA3-HA expression vector and primers used for the amplification are described in Supplementary Data 1. All

mutations and deletions were constructed with the Fast-mutagenesis Kit (TransGen Biotech, FM111-01), Quick Change Lightning Multi Site-Directed Mutagenesis Kit (Stratagene, 210513) or overlapping PCR strategy respectively. All constructs were confirmed by DNA sequencing. Antibody against human MAVS was raised by immunizing rabbits with recombinant protein His-sumo-hMAVS-(aa-301-460) and used at a dilution of 1:10,000. Commercial antibodies included anti-Flag (Sigma, F3165, F7425, dilution 1:5,000), anti-Tubulin (Sigma, T5168, dilution 1:7,500), anti-HA (Cell Signaling Technology, 3724 S, dilution 1:2,000), anti-TBK1 (Cell Signaling Technology, 3504, dilution 1:1,000), anti-pTBK1 S172 (Cell Signaling Technology, 5483 S, dilution 1:1,000), anti-IRF3 (Proteintech, 11312-1-AP, dilution 1:2,000), anti-TRAF3 (Santa Cruz, SC-1828, dilution 1:1,000), anti-Prohibitin (Abcam, ab75766, dilution 1:10,000), anti-RIG-I (Cell Signaling Technology, 3743, dilution 1:1,000), anti-WDR77 (Abcam, ab154190, 1:2,000), anti-pIRF3 S386 (Cell Signaling Technology, 37829, dilution 1:1,000), anti-PRMT5 (Abcam, ab109451, dilution 1:10,000), anti-MAVS (mouse) (Cell Signaling Technology, 4983 S, dilution 1:1,000), HRP anti-rabbit IgG (goat) (Promega, W4011, dilution 1:5,000), HRP anti-mouse IgG (goat) (Promega, W4021, dilution 1:5,000), Alexa Fluor™ 488 anti-rabbit IgG (goat) (Invitrogen, A11034, dilution 1:1,000), Cyanine Cy™3 anti-mouse IgG (goat) (Jackson ImmunoResearch, 115-165-146, dilution 1:1,000).

### Cells and viruses

HEK293T cells, HEK293 cells, HeLa cells, Vero cells, MEF cells, PEMs (isolated from C57BL/6 mouse) and BMDMs (isolated from C57BL/6 mouse and induced by M-CSF for 7 days) were cultured in DMEM supplemented with 10% fetal bovine serum (FBS, ExCell Bio, FSP500), penicillin (100 U/ml) and streptomycin (100 μg/ml). HEK293T (CRL-3216) and MEF (CRL-1658) were from American Type Culture Collection (ATCC). HEK293 (GNHu 43), HeLa (TCHu187) and Vero (SCSP-520) cells were purchased from the Cell Resource Center (Shanghai Institute of Biochemistry and Cell Biology). Sendai virus (Cantell strain) was provided by Dr. Xiaozhen Liang (Institute Pasteur of shanghai Chinese Academy of Sciences) and IAV (influenza A/Puerto Rico/8/1934 [H1N1] PR8 strain) was from Dr. Xiao Su (Institute Pasteur of Shanghai Chinese Academy of Sciences). Recombinant virus VSV-ΔM51-GFP and herpesvirus herpes simplex virus-1 (HSV-1) were amplified in Vero cells.

### Generation of knock-out cell lines based on HEK293T, HEK293, HeLa and MEF

*WDR77*[-/-] and *PRMT5*[-/-] knock-out cells were established by CRISPR/Cas9 technique and the guide RNA sequences are described in Supplemental Table 1. Briefly, guide RNAs were cloned into a CRISPR/Cas9-based vector pX330 with a puromycin-resistant selection marker. These vectors were transfected into HEK293T cells by Lipofectamine 3000 (Invitrogen, L3000015). After puromycin (0.5 μg/ml) selection, single colonies were picked and verified by DNA sequencing and immunoblotting.

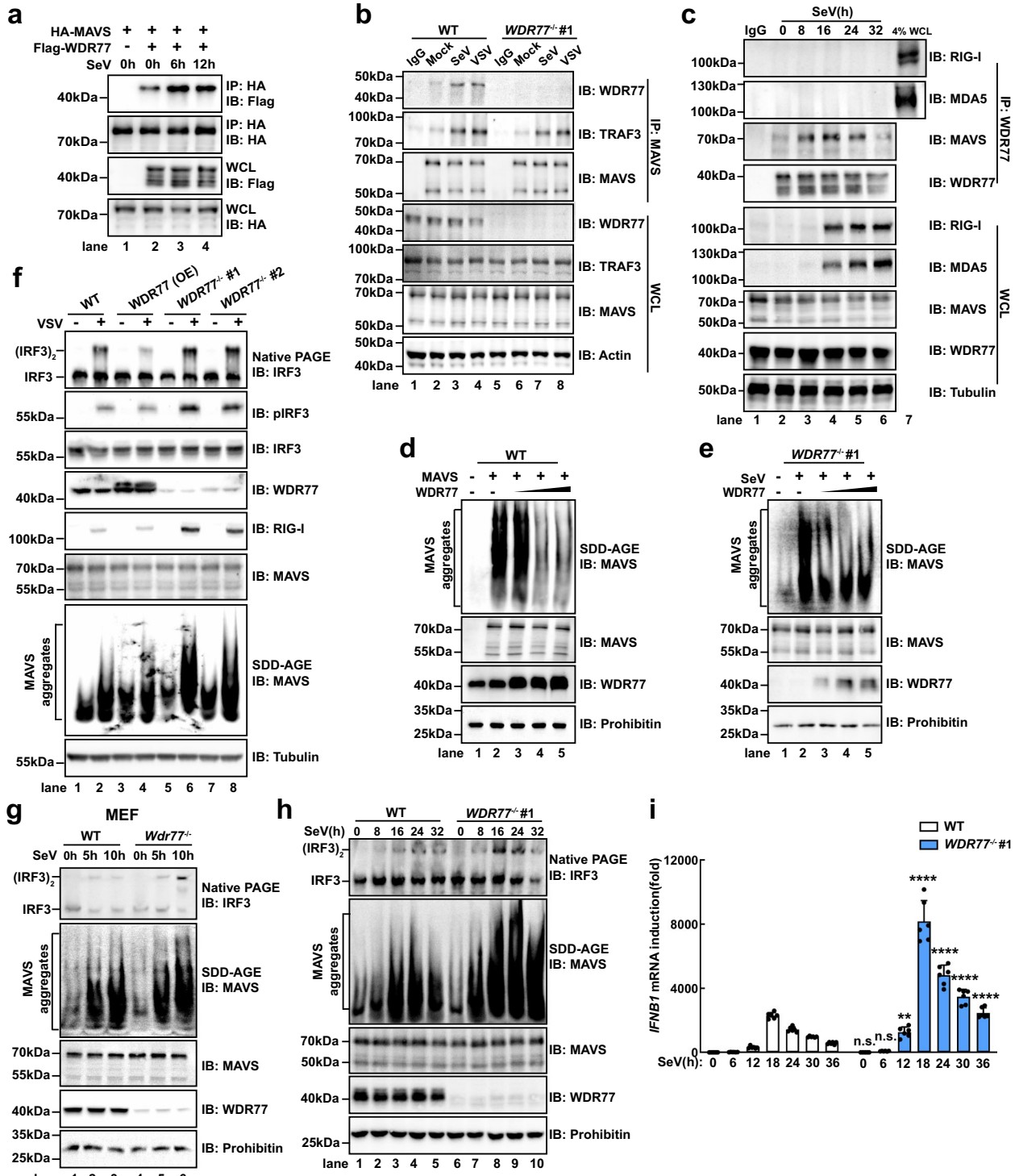

## Immunoprecipitations and immunoblot analysis

For immunoprecipitation, cells were harvested 36 h after transfection and lysed in IP buffer (HEPES 20 mM pH 7.5, 5 mM $MgCl_2$, KCl 10 mM, EGTA 0.5 mM, 1% Triton X-100 and protease inhibitor cocktail (Roche, 11697498001)). Following a brief centrifugation to clear up cellular debris, anti-FLAG M2 affinity gel (Sigma, A2220) or anti-HA agarose beads (Thermo, 26181) were added to cell lysate. After incubation for 4 h at 4 °C, FLAG or HA beads were spun down and washed with lysis buffer three times followed by immunoblotting analysis. For co-immunoprecipitation, cell lysates were collected and incubated with protein G Plus-Agrose Immunoprecipitation reagent (Santa Cruz Biotechnology, sc-2003) together with 1 μg of antibodies as indicated.

After incubation for 4 h, beads were washed with IP buffer five times. Immunoprecipitated products were boiled in sample buffer (Tris-HCl 60 mM pH 6.8, 1% SDS, 5% glycerol, 0.005% bromophenol blue and 1% β-mercaptoethanol) and subjected to SDS–PAGE followed by immunoblotting.

### Fluorescence microscopy
Wild type and $WDR77^{-/-}$ HEK293T cells were seeded in 6-well dish with a density of 60%. 12 h post-infection with VSV, fluorescent images were taken for live cells using Olympus IX71 inverted fluorescence microscope. To determine subcellular localization of WDR77, HEK293T cells were transfected with pcDNA3-HA-WDR77

**Fig. 4 | WDR77 inhibits MAVS aggregation and activation. a** HEK293T cells were co-transfected with Flag-WDR77 and HA-MAVS for 24 h followed by SeV stimulation. Cells were harvested and subjected to immunoprecipitation assay and immunoblotting. **b** WT or *WDR77*[-/-] #1 HEK293T cells were stimulated with or without SeV or VSV for 12 h. Cells were collected and subjected to immunoprecipitation assay and immunoblotting. **c** HEK293T cells were stimulated with SeV and harvested at various time points as indicated for immunoprecipitation assay and immunoblotting. The sample shown in lane 7 was 4% of input (whole cell lysate (WCL) harvested 32 h post infection. **d** HEK293T cells were transfected with increasing amounts of Flag-WDR77-expressing plasmids for 24 h, and then transfected with Flag-MAVS plasmids for 12 h. Cells were harvested for subcellular fractionation. P5 fractions were subjected to SDD-AGE to examine MAVS aggregation. **e** *WDR77*[-/-] #1 cells were transfected with increasing amounts of Flag-WDR77-expressing plasmids for 24 h, and then stimulated with or without SeV for 8 h. Cells were harvested for subcellular fractionation. P5 fractions were subjected to SDD-AGE to examine MAVS aggregation. **f** WT and *WDR77*[-/-] HEK293T cells transfected

with or without Flag-WDR77 were stimulated with or without VSV for 8 h. Cells were harvested for subcellular fractionation. S5 fractions were subjected to native PAGE to examine IRF3 dimer, and P5 fractions were subjected to SDD-AGE to examine MAVS aggregation. WCLs were subjected to SDS-PAGE to examine IRF3 phosphorylation. **g** WT or *Wdr77*[-/-] MEF cells were stimulated with SeV for indicated times. Cells were harvested for subcellular fractionation. S5 fractions were subjected to native PAGE to examine IRF3 dimer, and P5 fractions were subjected to SDD-AGE to examine MAVS aggregation. **h** WT or *WDR77*[-/-] #1 HEK293T cells were stimulated with SeV. Cells were harvested at specified time points post infection and subjected to immunoblotting. **i** WT or *WDR77*[-/-] #1 HEK293T cells were stimulated with SeV. Cells were harvested at specified time points post infection and *IFNB1* induction was measured by qPCR (*IFNB1*: all [ns]$p > 0.9999$, ****$p < 0.0001$, **$p = 0.0018$). Data are representative of two independent experiments with similar results (**a–h**), or three independent experiments (**i**) (mean ± SD of three biological replicates with 2 technical replicates). *P*-values were determined by two-way ANOVA (Šídák's test) (**i**). Source data are provided as a Source Data file.

and Flag-MAVS by Lipofectamine 3000. Twenty-four hours after transfection, cells were infected with or without SeV for 8 h. Cells were then washed with pre-warmed 1×PBS, fixed with 4% formaldehyde for 15 min, and permeabilized with 0.5% Triton X-100 for 15 min. After blocking with 3% bovine serum albumin (Roche,10735078001) at room temperature for 1 h, the cells were incubated with anti-HA or anti-Flag antibodies respectively overnight at 4 °C. After incubation, cells were washed with pre-warmed 1×PBS three times. Before fluorescent imaging, cells were incubated with secondary antibodies as indicated and 1 μg/ml DAPI (Beyotime, C1002). All the images were taken with Leica TCS SP8 STED X laser scanning confocal microscope.

## Generation of knock-down cell lines by shRNA
Sequences of shRNAs targeting WDR77 and PRMT5 are described in Supplementary Data 1. HEK293T WT cells were seeded in 6-well plates and transfected with shRNA-harboring vector pLKO.1 and viral packaging vectors using Lipofectamine 2000 (Invitrogen, 11668019). Culture medium containing packaged lentiviral particles was collected after 48 h. HEK293T cells were infected with lentiviral particles for 24 h and treated with puromycin for 48 h. Cells were then harvested for various stimulations. Stable knockdown cell lines were obtained by select single colony, which was verified by DNA sequencing and immunoblotting.

## Luciferase reporter assay
HEK293T cells were seeded in 12-well plates at a density of $1 \times 10^5$ cells per well and transfected with 20 ng of reporter gene (*IFNB1*-luciferase, *ISRE*-luciferase or *NFKB1*-luciferase), 20 ng of pCMV-LacZ, and expression vectors as indicated by Lipofectamine 3000. Twenty-four hours post-transfection, cells were stimulated with or without VSV or SeV. After cells were harvested and lysed in Passive Lysis Buffer (Promega, E1941), firefly luciferase activities were measured with a luminometer using the Luciferase Reporter Kit (Promega, E2710) and LacZ activities were measured by o-Nitrophenyl-b-D-Galactopyranoside (ONPG) assay following a protocol provided by Sigma Technical Bulletin (GALA-1KT). Induction of firefly luciferase was normalized to LacZ activity. Data were shown as induction fold over controls transfected with empty vectors.

## Quantitative PCR
Total RNA was extracted from cells using Trizol (Tiangen, 4992730), and the reverse transcription was performed using HiScript® III SuperMix for qPCR (+gDNA wiper) (Vazyme, R323-01). cDNAs were then used as templates for qPCR assay on Roche Applied Science LightCycler 480 with ChamQ Universal SYBR qPCR Master Mix (Vazyme, Q711-02/03). The induction fold was determined with the ΔΔCq method and qPCR primers used to amplify specific genes were listed in Supplementary Data 1.

## ELISA
Concentrations of the IFN-β in cell culture medium and mouse serum were measured by Human Interferon β ELISA kit (Cusabio, CSB-E09889h) or Mouse Interferon β ELISA kit (Cusabio, CSB-E04945m) according to the manufacturer's instructions.

## Expression and purification of recombinant proteins
The prokaryotic expression vector pET-28a-His-Sumo-MAVS-(aa-1-510) and pET-28a-His-MBP-WDR77 were transformed in BL21 strain (TransGen Biotech). Expression of recombinant proteins were induced by 0.2 mM isopropyl-b-Dthiogalactoside (IPTG) at 18 °C for 4 h. After sonication in lysis buffer (Tris-Cl 10 mM pH 8.0, NaCl 0.5 M, β-mercaptoethanol 5 mM, dithiothreitol 0.5 mM, 5% Glycerol and phenylmethylsulphonyl fluoride 0.5 mM), cell lysates were centrifuged at 18,000 g for 30 min. Recombinant proteins Sumo-MAVS-(aa-1-510) and MBP-WDR77 were purified with Ni-NTA agarose beads according to the manufacturer's protocol (Qiagen, 30210). Then His6-Sumo-MAVS were loaded onto a Hitrap-Q column, and eluted with Buffer A (Tris 10 mM pH 7.5, 5% glycerol, DTT 2 mM, EDTA 1 mM, PMSF 0.5 mM, and AEBSF0.5 mM) containing a gradient of NaCl from 0.1 M to 0.5 M. The fractions containing His6-Sumo-MAVS, which was eluted by about 300 mM NaCl, were pooled and loaded onto a Superdex-200 gel filtration column equilibrated with Buffer B (Tris-HCl 10 mM pH 8.0, NaCl 150 mM, DTT 1 mM, EDTA 1 mM, PMSF 0.5 mM, and AEBSF 0.5 mM). ÄKTA micro (GE Healthcare) and a 24 ml Superdex-200 were used for large-scale purification.

## Subcellular fractionation
HEK293T or MEF cells were homogenized in hypotonic buffer (Tris-Cl 10 mM pH 7.5, KCl 10 mM, EGTA 0.5 mM and MgCl$_2$ 1.5 mM) by douncing. After centrifugation at 1,000 g for 10 min at 4 °C, supernatant (S1) and pellet (P1) were obtained. The S1 was further centrifuged at 10,000 g for 10 min at 4 °C, the supernatant was S5 fraction and the pellet was P5 fraction. P5 and S5 were used for IRF3 dimerization assay in vitro followed by native PAGE and immunoblotting as described.

## Native PAGE and SDD-AGE
Native PAGE was employed to analyze IRF3 dimerization as described previously[51]. In brief, the samples were mixed with 5×native loading buffer (Tris-HCl 100 mM pH 8.8, 50% Glycerol, 0.04% Bromophenol Blue), and loaded onto a 9% polyacrylamide gel without SDS, which was pre-run in the native running buffer (Tris-HCl 25 mM pH 8.8, Glycine 19.2 mM, 0.4% Doc-Na) for 60 min with a constant voltage of 200 V at 4 °C. After electrophoresis for another 60 min, immunoblotting was performed. SDD-AGE was used to analyze MAVS aggregation. A vertical 1.5% agarose gel was prepared and crude mitochondrial fractions (P5) were obtained by differential

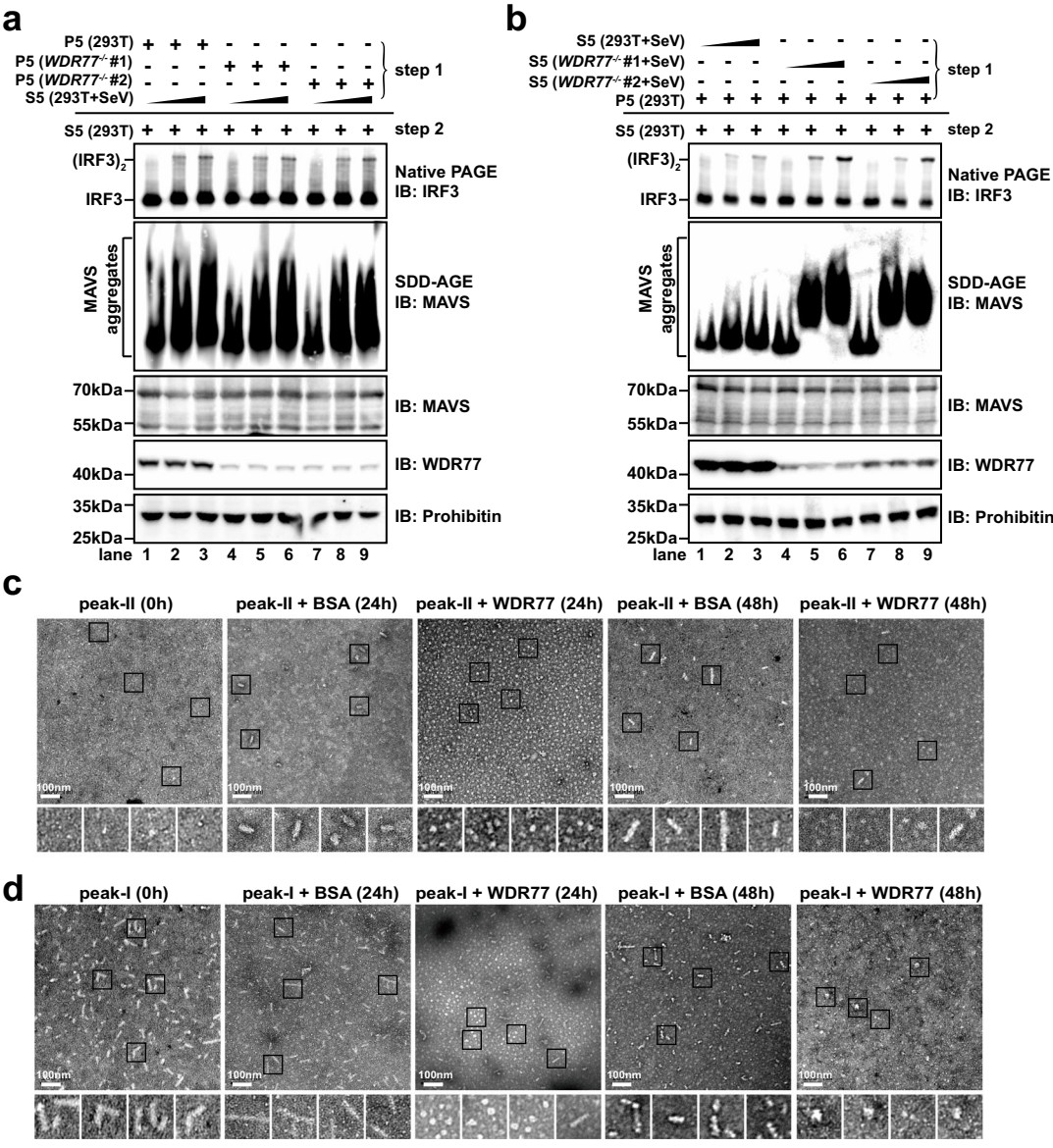

**Fig. 5 | WDR77 impedes MAVS to form prion-like filaments. a** A two-step assay was performed. In step 1, HEK293T cells were infected with SeV for 8 h and harvested to get S5 fraction. Increasing amounts of S5 fractions were incubated with P5 fractions from WT or *WDR77*$^{-/-}$ HEK293T cells at 30 °C for 1 h. P5 fractions were then isolated. In step 2, P5 fractions from step 1 were incubated with S5 fractions from WT HEK293T cells at 30 °C for another 1 h. IRF3 dimerization was then analyzed by native PAGE and MAVS aggregation was analyzed by SDD-AGE. Blots shown are representative of *n* = 2 biological replicates. **b** A two-step assay was performed. In step 1, S5 fractions were prepared from WT and *WDR77*$^{-/-}$ HEK293T cells infected with SeV for 8 h, which were incubated with P5 fractions from WT HEK293T cells at 30 °C for 1 h. P5 fractions were then isolated. In step 2, P5 fractions from step 1 were incubated with S5 fractions from WT HEK293T cells at 30 °C for another 1 h. IRF3 dimerization was analyzed by native PAGE and MAVS aggregation was analyzed by SDD-AGE. Blots shown are representative of *n* = 2 biological replicates. **c** Recombinant MAVS was separated into two peaks by gel filtration chromatography, i.e., Peak-I and Peak-II. MAVS (Peak-II) was incubated with BSA or recombinant WDR77 at 4 °C. At various time points, the mixtures were subjected to negative staining and examination under electron microscopy. Scale bars indicate 100 nm. Representative images of *n* = 3. **d** MAVS (Peak-I) was incubated with BSA or recombinant WDR77 at 4 °C. At various time points, the mixtures were subjected to negative staining and examination under electron microscopy. Scale bars indicate 100 nm. Data are representative of two independent experiments with similar results (**a**, **b**), or three independent experiments with similar results (**c**, **d**). Source data are provided as a Source Data file.

centrifugation. P5 was resuspended in 1×SDD loading (10% Glycerol, 0.5×TBE, 2% SDS, and 0.0025% Bromophenol Blue) and loaded onto the agarose gel. Electrophoresis in the running buffer (0.5×TBE, 0.1% SDS) was executed for 40 min with a constant voltage of 100 V at 4 °C, followed by immunoblotting analysis.

**Electron microscopy**
Copper grids (Ted Pella Inc.) coated with a layer of thin carbon film (5-10 nm) were rendered hydrophilic by glow-discharge in air. Seven microliters of purified MAVS samples were loaded onto the grids, incubated for 3 min, and stained with 2% uranyl acetate. After air

drying for a few minutes, the grids were examined under a FEI Tecnai G2 Spirit 120 KV electron microscope. Images were taken in low-dose mode (20 electron/Å 2) at 67,000 x with defocus levels varying between 0.5 to 1.2 microns. Regular calibrations of the microscope magnifications found that the nominal 67,000x used here was always accurate within 3.0% error.

**Plaque assay**
Plaque assay was performed as described[38]. In brief, Vero cells in 6-well plates were infected with serial dilutions of VSV for 1 h. Serum-free medium was removed and replaced with MEM medium containing 1%

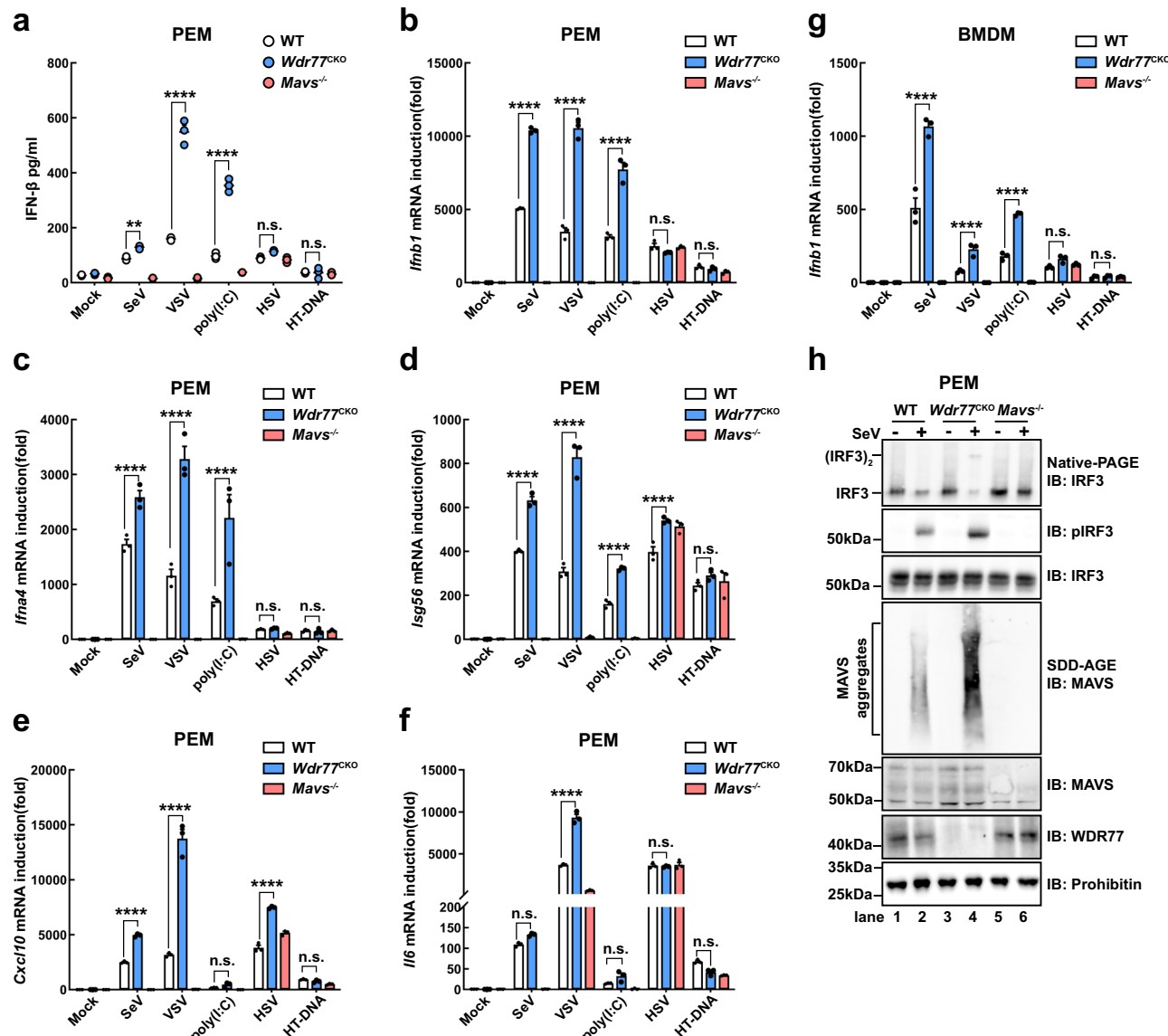

**Fig. 6 | WDR77 dampens antiviral immune response in mouse primary cells.**
**a–f** PEMs from WT, *Wdr77*CKO and *Mavs*-/- mice respectively were collected. PEMs were then stimulated for 6 h with or without VSV, SeV, poly(I:C), HSV or HT-DNA as indicated. Culture mediums were then collected and IFN-β was measured by ELISA (**a**). *Ifnb1* (**b**), *Ifna4* (**c**), *Isg56* (**d**), *Cxcl10* (**e**), *Il-6* (**f**) induction were measured by qPCR (For **a**, IFN-β: **\*\***$p = 0.0021$ **\*\*\*\***$p < 0.0001$, **\*\*\*\***$p < 0.0001$, ns$p = 0.0790$, ns$p = 0.9993$ in sequence; for **b**, *Ifnb1*: all **\*\*\*\***$p < 0.0001$, ns$p = 0.1813$, ns$p = 0.8208$ in sequence; for **c**, *Ifna4*: all **\*\*\*\***$p < 0.0001$, ns$p = 0.9997$, ns$p = 0.9998$ in sequence; for **d**, *Isg56*: all **\*\*\*\***$p < 0.0001$, ns$p = 0.1378$; For **e**, *Cxcl10*: all **\*\*\*\***$p < 0.0001$, ns$p = 0.5755$, ns$p = 0.7986$ in sequence; for **f**, *Il-6*: ns$p = 0.9872$, **\*\*\*\***$p < 0.0001$, ns$p = 0.9932$, ns$p = 0.7531$, ns$p = 0.9849$ in sequence). **g** BMDMs from WT, *Wdr77*CKO and *Mavs*-/- mice respectively were collected. BMDMs were then stimulated for 6 h with or without VSV, SeV, poly(I:C), HSV or HT-DNA as indicated. *Ifnb1* induction was measured by qPCR. (All **\*\*\*\***$p < 0.0001$, ns$p = 0.1224$, ns$p = 0.9878$ in sequence). **h** PEMs from WT, *Wdr77*CKO and *Mavs*-/- mice were collected respectively. PEMs were infected with or without SeV for 6 h. Cells were collected for subcellular fractionation, S5 fractions were subjected to native PAGE to examine IRF3 dimer. P5 fractions were subjected to SDD-AGE to examine MAVS aggregation and WCL were subjected to SDS-PAGE to examine IRF3 phosphorylation. Data are representative of two independent experiments with similar results (**h**), or three independent experiments (**a**–**g**) (mean ± SEM of three biological replicates). *P*-values were determined by two-way ANOVA (Tukey's test) (**a**, **b**, **g**), two-way ANOVA (Šídák's test) (**c**) or two-way ANOVA (Dunnett's test) (**d**–**f**). n.s. indicates no statistical significance. Source data are provided as a Source Data file.

low melting temperature agarose (Lonza) and 10% FBS, followed by incubation for 48 h at 37 °C. The wells were stained with 0.1% crystal violet solution for 5 min at room temperature, and the number of plaques in the plate was counted.

## Statistical analysis
Data are presented as mean value ± SD (standard deviation) or ± SEM (standard error of the mean) of three independent experiments unless otherwise stated. Statistical analyses were performed with GraphPad Prism 9.0.0 software. Unpaired two-sided Student's *t*-test, ordinary one-way ANOVA and two-way ANOVA were used to analyze the significance of the differences between groups. Survival data of mice were analyzed using Gehan-Breslow-Wilcoxon test. $P < 0.05$ was considered statistically significant and n.s. indicates no statistical significance ($P > 0.05$).

## Reporting summary
Further information on research design is available in the Nature Portfolio Reporting Summary linked to this article.

## Data availability
A reporting summary for this article is available as a supplementary file. The main data supporting the findings of this study are available within

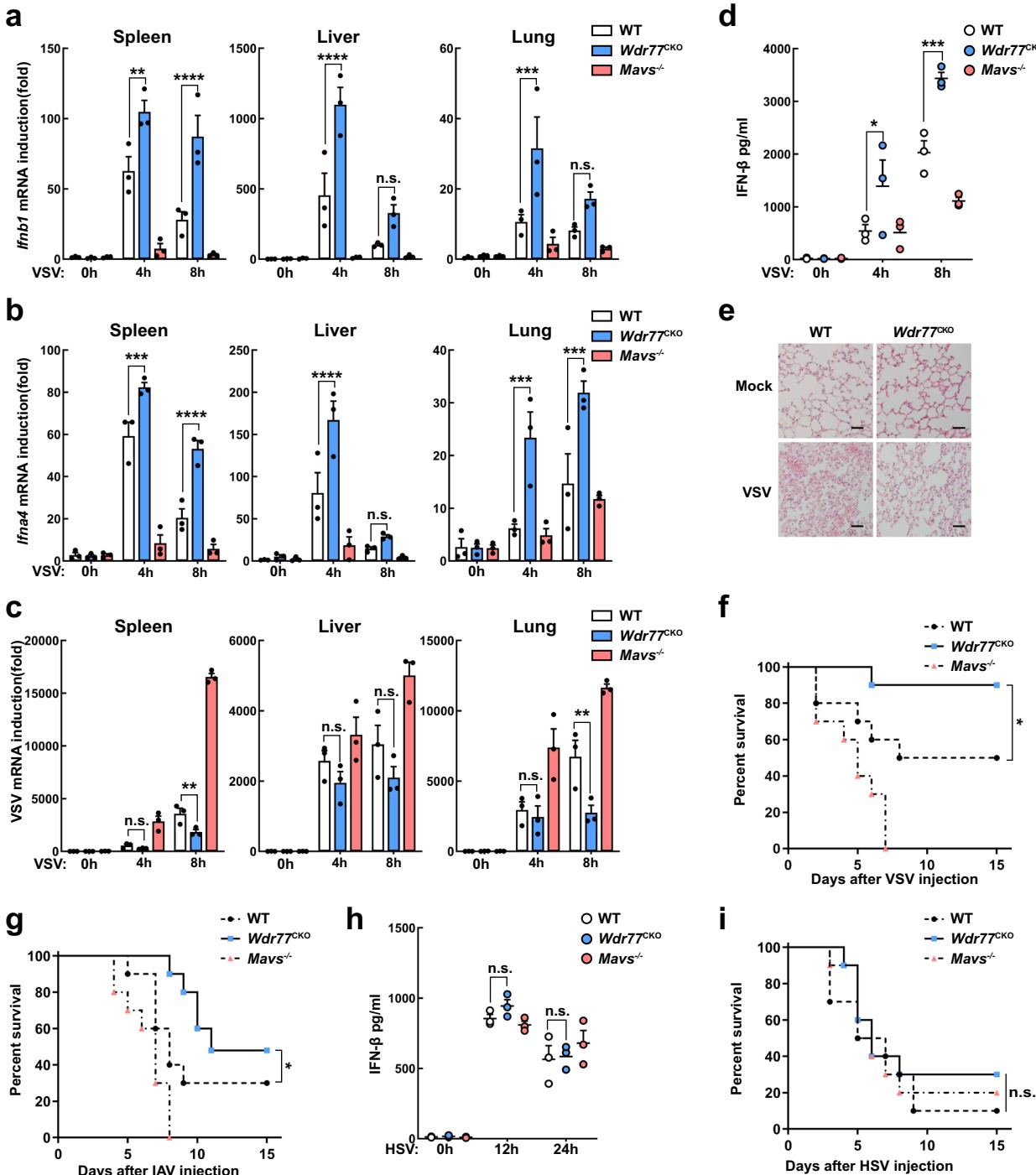

**Fig. 7 | WDR77 negatively regulates antiviral innate immunity in mice. a–d** WT, *Wdr77*^CKO and *Mavs*^-/- mice (*n* = 3 each) were infected intravenously with VSV at 2 × 10[7] PFU per mouse. The spleens, livers, and lungs were collected 4 h or 8 h after infection. *Ifnb1* (**a**), *Ifna4* (**b**) inductions and VSV RNA levels (**c**) were measured respectively by qPCR. The sera were collected and used for the measurement of IFN-β by ELISA (**d**) (For **a**, *Ifnb1*: **p = 0.0013, ****p < 0.0001, ****p < 0.0001, ^ns p = 0.0815, ***p = 0.0006, ^ns p = 0.1397 in sequence; for **b**, *Ifna4*: ***p = 0.0004, ****p < 0.0001, ****p < 0.0001, ^ns p = 0.6194, ***p = 0.0008, ***p = 0.0008 in sequence; For **c**, VSV: ^ns p = 0.7616, **p = 0.0010, ^ns p = 0.3889, ^ns p = 0.1316, ^ns p = 0.8740, **p = 0.0023 in sequence; for **d**, IFN-β: * p = 0.0216, ***p = 0.0003 in sequence). **e** Hematoxylin-eosin staining of lung sections from WT and *Wdr77*^CKO mice infected intravenously with VSV at 2 × 10[7] PFU per mouse for 12 h. Scale bars indicate 50 μm. **f** WT, *Wdr77*^CKO and *Mavs*^-/- mice (*n* = 10 each) were injected intravenously with VSV at 5 × 10[7] PFU per mouse, and the survival rates were monitored

for 15 days (*p = 0.049). **g** WT, *Wdr77*^CKO and *Mavs*^-/- mice (*n* = 10 each) were injected intranasally with IAV at 8 × 10[2] PFU per mouse, and the survival rates were monitored for 15 days (*p = 0.045). **h** WT, *Wdr77*^CKO and *Mavs*^-/- mice (*n* = 3 each) were infected intravenously with HSV at 1.5 × 10[8] PFU per mouse. The sera were collected 12 h or 24 h after infection and used for the measurement of IFN-β by ELISA (IFN-β: ^ns p = 0.5412, ^ns p = 0.9892 in sequence). **i** WT, *Wdr77*^CKO and *Mavs*^-/- mice (*n* = 10 each) were injected intravenously with HSV at 1.5 × 10[8] PFU per mouse, and the survival rates were monitored for 15 days (^ns p = 0.445). Data are representative of three independent experiments with similar results (**e**), or three independent experiments (**a–d**, and **h**) (mean ± SEM of three biological replicates). *P*-values were determined by two-way ANOVA (Tukey's test) (**a**, **c**), two-way ANOVA (Šídák's test) (**b**, **d**, **h**) or Gehan-Breslow-Wilcoxon test (**f**, **g**, **i**). n.s. indicates no statistical significance. Source data are provided as a Source Data file.

the article and its Supplementary Figures. The source data underlying Figs. 1–7, Supplementary Figs. 1–8 are provided as a Source Data file. In this study, the identification of WDR77 was accomplished through mass spectrometry analysis. Furthermore, the analysis also revealed other MAVS-associated proteins; however, as they are not directly pertinent to the focus of this study, they were not included in the manuscript. Nonetheless, additional information regarding these identified proteins can be provided upon reasonable request. Source data are provided with this paper.

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

## Acknowledgements

This work was supported by grants from Shanghai Institute of Biochemistry and Cell Biology, the Chinese Academy of Sciences Pilot Strategic Science and Technology Project B (XDB37030207 to F.H.), the National Natural Science Foundation of China (32070902 to F.H.), the National Key Research and Development Program of China (No. 2020YFA0804200 to H.Y.) and NSFC grant (No. 82073166, 82273203 to H.Y.). We thank Dr. Bofeng Yuan for her pilot experiments on the study. We thank Dr. Hongyan Wang (SIBCB) for providing Lyz2-Cre mice.

## Author contributions

J.L. conducted experiments. M.R., Q.W., Y.J., J.Z., S.Q. and C.W. helped with experiments on mice. M.R., R.Z., Y.D. and X.H. provided technical help. L.L. and S.C. did mass spectrometric analysis. F.H. and J.L. organized the study and prepared the manuscript. H.Y. provided intellectual inputs and edited the manuscript. All authors discussed the results and commented on the manuscript.

## Competing interests

The authors declare no competing interests.
