## [Peer Review File · Nature Communications]

WDR77 inhibits prion-like aggregation of MAVS to limit antiviral innate immune responseREVIEWER COMMENTS

Reviewer #1 (Remarks to the Author):

In this manuscript, Li et al., identify that WDR77 is a potential antagonist to MAVS aggregation and subsequent downstream signaling in the activation of type-I IFN. They demonstrate that loss of WDR77 potentiates antiviral response and limits viral replication. They further demonstrate that it physically interacts with MAVS and prevents its aggregation and propose that through this functional consequence, WDR77 is able to mute the antiviral signaling mediated through MAVS. They have demonstrated these results using human cell lines and also mouse models in the process of arriving at their conclusion. Prima facie, the results are interesting and original, and identify a new mechanism of negative regulation of MAVS mediated IFN production. Though these results are very interesting to a vast reader base, the authors need to provide additional information before the manuscript being acceptable for publication.

Major comments

1. In Figure 1 and throughout the manuscript, IFN- β production was measured either by luciferase assay or by RT-PCR. Since these are not absolute measures of total regulation, these assays should be supported by ELISA.
2. In Figure 2, IFN β activation by VSV was more profound in PRMT5 KD than WDR KD. Please explain the possible reasons, especially in the light that the former molecule was not identified a potential molecule of interest in the later part of the study.
3. The VSV infectious titer assay in WDR-/- 293T cells only shows a drop by three folds, which is not a very significant one. Normally, in MAVS deficient cells, the viral titers increase multiple logs. This suggests that it only has a limited control over MAVS functions and hence may not be an important player in its regulation.
4. In Figure 3, S396D IRF3-induced ISRE activation of ISRE luciferase was significantly smaller in comparison with the other molecules in WDR77 KO cell background while the activation was robust with WDR77 transfection in 293T cells (Fig 3b). Please explain the significance in this differential response in the larger context of RLR signalling.
5. In Figure 4, TRAF3 was found associating with both WDR77 and MAVS, independently. What is the significance of TRAF3 in this association if they bind independently to both MAVS and WDR77? There does not seem to be any follow up on this in the subsequent parts, nor there is any studies on TRAF2. This seems puzzling.
6. The section title in line 260 reads, "Wdr77 is essential for antiviral immune response in mouse primary cells". How is Wdr77 essential for antiviral response when it is antagonist to the response itself?
7. Line 272 reads "PEMs were prepared from WT, Wdr77CKO and Mavs-/- mice followed by various treatments as indicated (Fig. 6a)." This statement seems to be wrong as the legend suggests that infections were performed in isolated cells, and not in animal after which the cells were isolated.
8. Though WDR77 was demonstrated as an antagonist to MAVS, how critical is WDR77 binding in inactivating MAVS during late stages of viral infection? There is no experimental set up that differentiates the dynamics of MAVS activation and inactivation during the whole process of infection, particularly between the early and the late stages of infection. It is merely proposed that WDR77 could be involved in the inactivation of MAVS signalling. However, more details on its temporal engagement with MAVS are required in order to consider it as an important player in muting MAVS signalling
9. What is the proposed mechanism by which WDR77 mutes MAVS signalling? Does WDR77 cause any PTM? how is its mode of action? Do WDR77 levels undergo dynamic changes during infection in order to inactivate MAVS?
10. Though Wdr CKO confirmed that its loss indeed results in lower viral titers, it does not rule out the possibility that its effect is independent of MAVS. To further demonstrate that WDR77 imparts suppression of MAVS mediated antiviral signalling, these studies should be tested in WdrCKO + MAVS KO double KO.

Minor comments

1. Certain statements on WDR77 have been confusing at places. For eg., line 136 states that "To further validate its antiviral function, three WDR77 knockout HEK293T cell lines...". While WDR77 is antagonistic to antiviral response and hence is proviral in function, it is

refereed to as "antiviral". This needs to be addressed throughout the manuscript.

2. Line 104-107 : Figure 1C where WDR77 mw is 40kDa except that WDR77 molecular weight is close to 45kDa. Discuss the possible reason for this discrepancy.

3. Line 208 states "Based on these data, we speculate that WDR77 might be involved in the downregulation of MAVS activity after an effective innate immune response is launched to avoid harmful inflammation". However, there is no immunoblot or any other data to show the status of viral infection and at what time WRD77 shows its maximum activity.

4. Line 307 states "These data indicate that WDR77 is an important negative regulator of the antiviral immune response against RNA virus in vivo". The authors have worked solely with VSV and SeV which are negative sense RNA viruses. Similar studies performed on a positive sense RNA virus can add further value to this study.

5. Line 334 states "Moreover, WDR77 deficiency resulted in enhanced activation of TBK1/IRF3 and upregulation of IFN- β induction in HEK293T, MEF cells and HeLa cells upon RNA virus but not DNA virus infection". Since this was not tested using a DNA virus but only with HT-DNA as a stimulant, the authors need to change this statement.

6. Line 710-713 : What is the duration of VSV infection in Figure 2g? The figure displays 12h while the legend shows 8h.

7. Line 713-714; Figure 2h-j. Were these performed in 293 as mentioned in the figure or 293T cells? There is no mention about HEK293 in material method. KO cell lines generation also mention HEK293T, HeLa and MEF only. If it is HEK293 then it has to be mentioned in cells and KO generation section in material and methods.

Reviewer #2 (Remarks to the Author):

This manuscript by Li et al describes a negative regulation mechanism of MAVS signaling by WDR77. Through affinity pull-down the authors identified WDR77 as a protein that interacted with MAVS and describe negative regulation of MAVS-mediated antiviral signaling through the inhibition of MAVS aggregation. The authors followed up these in vitro results using a myeloid-specific WDR77 knockout mice to show enhanced resistance to VSV and IAV, but not HSV. The results convincingly show that WDR77 is indeed a negative regulator of the RIG-I-MAVS signaling, but the molecular mechanism that the authors propose are not always convincing and confusing. They claim that WDR77 is recruited to MAVS following activation of RLR signaling (pIC or SeV treatment) Fig. 3 and 4. What is not clear from the results, how or what causes the stimulation of WDR77. Again, this model is inconsistent with results from Fig. 1 where expression and purification of HA-MAVS from unstimulated cells led to its identification as a MAVS interacting protein. Following are some suggestions for further experiments and major revision of this manuscript.

1. Major editing of the manuscript is needed to make it clear and understandable.

2. The manuscript focuses extensively on detecting MAVS aggregation and how it is inhibited by WDR77. However, some of these assays are highly qualitative such as AGE, and EM studies that are difficult to interpret without further supporting evidence. For example, in the EM studies it is not clear how are MAVS in peak 1 and peak 2 are dis-aggregating or aggregating spontaneously.

3. Since RIG-I or MDA5 are the only known RNA sensors in this context, they might be involved in upstream sensing and initiating WDR77 recruitment to MAVS. This should be investigated.

Alternatively, if WDR77 is pre-associated with MAVS, and help resolve the MAVS aggregates upon stimulation, it should be clearly demonstrated and explained.

4. Another relevant question in this context will be if WDR77 is a downstream target of MAVS signaling. In this scenario, it is conceivable that during the late stage of MAVS signaling WDR77 is transcribed, accumulated, and resolves MAVS aggregates to terminate the activation of RLR signaling.

Reviewer #3 (Remarks to the Author):

The manuscript by Li and colleagues from the Shanghai Institute of Molecular & Cellular Biology identified WDR77 as a MAVS-associating protein following a genetic screen using extract from mitochondria. WDR77 was shown to bind to the MAVS proline-rich region through its WD2-WD3-WD4 domain, and decreases RIG-I dependent signaling to the antiviral response. The data illustrates that WDR77 is recruited to MAVS to disassemble the formation of prion-like aggregates of MAVS. WDR77 deficiency generated by CRISP/R-cas results in a phenotype that potentiates induction of antiviral genes after VSV infection. Myeloid specific WDR77-deficient mice are more resistant to RNA virus infection.

It is difficult to identify a set of experiments that have not been completed in this study; many approaches have been taken – co-transfections to measure IFN response, challenge with VSV in the presence or absence of WDR77, knock-down of WDR77 or knock-out by CRISP/R-cas targeting are all included. The results argue that WDR77 is involved in termination of the antiviral response by dissociating the prion-like MAVS aggregates. The experiments are generally well performed and the English grammar and style are good – another round of proof-reading will improve the text. Several points for clarification:

1. Authors find only two proteins interacting with MAVS – WDR77 and PRMT5; could the authors discuss the reasons for this surprising finding – only 2 proteins interacting with MAVS. In fact many associated proteins have been described.
2. The authors imply a temporal and perhaps stochastic model of regulation - that is WDR77 appears/binds to MAVS later in infection to dissociate MAVS. Any evidence that WDR77 is induced at late times ? and in vivo is the amount of WDR sufficient to mediate these effects ?
3. Throughout the manuscript, authors refer to the decrease in antiviral response as 'significantly increased' or 'dramatically decreased' but in fact the effects in down modulating antiviral activity is rather modest ~3 fold throughout. Is such a modest effect sufficient to block the complete antiviral response, or is WDR mechanism one of several interactions that contribute to MAVS dissociation?
4. Human WD repeat domain 77 (WDR77, also MEP50) protein is a subunit of the 20S methylosome complex that includes type II protein arginine methyltransferase 5 (PRMT5), an enzyme responsible for mono- and symmetric dimethylation of arginine . The data does not clarify between a model in which WDR binds to MAVS vs a mechanism that is still dependent on PRMT5 activity, but in association with WDR77. A knockdown of PRMT5, or co-expression of both molecules, or point mutations of PRMT5 should help to clarify this point. Authors have likely performed such experiments. Essentially the authors have previously argued that MAVS regulation is dependent on PRMT7 activity and now argue that regulation is dependent on WDR77 without involvement of PRMT5.

The observations of the manuscript are interesting but without a clarification of mechanism, the study is of limited significance, particularly since the authors previously argued for a different mechanism of PRMT action on MAVS.

Response to reviewer's remarks

We appreciate the reviewers' critical comments and suggestions. We have made extensive and careful revisions accordingly and our point-by-point response is provided.

Reviewer #1

Reviewer #1 (Remarks to the Author):

In this manuscript, Li et al., identify that WDR77 is a potential antagonist to MAVS aggregation and subsequent downstream signaling in the activation of type-I IFN. They demonstrate that loss of WDR77 potentiates antiviral response and limits viral replication. They further demonstrate that it physically interacts with MAVS and prevents its aggregation and propose that through this functional consequence, WDR77 is able to mute the antiviral signaling mediated through MAVS. They have demonstrated these results using human cell lines and also mouse models in the process of arriving at their conclusion. Prima facie, the results are interesting and original, and identify a new mechanism of negative regulation of MAVS mediated IFN production. Though these results are very interesting to a vast reader base, the authors need to provide additional information before the manuscript being acceptable for publication.

Major comments:

1. In Figure 1 and throughout the manuscript, IFN- β production was measured either by luciferase assay or by RT-PCR. Since these are not absolute measures of total regulation, these assays should be supported by ELISA.

Reply: We thank the reviewer for these suggestions. We now provide ELISA data for IFN- β production in our revised manuscript (**Fig. 1h, k, Page 35; Supplementary Fig. 3b, Page 55; Supplementary Fig. 4h, Page 57**).

2. In Figure 2, IFNB activation by VSV was more profound in PRMT5 KD than WDR KD. Please explain the possible reasons, especially in the light that the former molecule was not identified a potential molecule of interest in the later part of the study.

Reply: We thank the reviewer for the comment. In Supplementary Fig 2c, IFNB activation by VSV was indeed more profound than PRMT5 KD. One of our explanations is that the expression of WDR77 and PRMT5 appears to be mutually dependent on each other, thereby in PRMT5 KD cells, WDR77 level is low as well. The other explanation is that in Supplementary Fig 2c, either PRMT5 or WDR77 was knocked down transiently, and a mixture of cells was assayed, which might complicate the data due to cellular heterogeneity.

To get clean and more reliable data, we generated stable knockdown cell lines derived from single colonies respectively (**Supplementary Fig. 2d, e, Page 53**). IFNB induction in these cells was examined, which showed that more IFNB was induced in WDR77 KD cells than in PRMT5 KD cells

(Supplementary Fig. 2f, g, Page 53).

Supplementary Fig. 2d

Supplementary Fig. 2f

Supplementary Fig. 2e

Supplementary Fig. 2g

3. The VSV infectious titer assay in WDR^{-/-} 293T cells only shows a drop by three folds, which is not a very significant one. Normally, in MAVS deficient cells, the viral titers increase multiple logs. This suggests that it only has a limited control over MAVS functions and hence may not be an important player in its regulation.

Reply: We thank the reviewer for the critical comment. Virus infectious titer may vary dependent on factors such as cell types used, virus strain and virus MOI employed in a specific experiment setting etc. To get a direct comparison of WDR77 and MAVS, we then examined VSV titer in *WDR77^{-/-}* as well as *MAVS^{-/-}* cells side by side. Our data showed that under the experimental conditions as described (**Lines 747-752 Page 39**), VSV infectious titer from *MAVS^{-/-}* cells was about four folds of that from WT cells. VSV titer from WT cells was approximately nine folds of that from *WDR77^{-/-}* cells (**Fig. 2g, h, Page 38**), suggesting that *WDR77^{-/-}* cells are much more resistant to VSV proliferation. These data indicate that WDR77 plays an important role in modulating MAVS function.

4. *In Figure 3, S396D IRF3-induced ISRE activation of ISRE luciferase was significantly smaller in comparison with the other molecules in WDR77 KO cell background while the activation was robust with WDR77 transfection in 293T cells (Fg 3b). Please explain the significance in this differential response in the larger context of RLR signalling.*

Reply: We thank the reviewer for the critical comment on our data. The magnitude in IRF3(S396D)-induced *ISRE* activation could be contingent on multiple factors, such as parental cells used, cell density and most importantly, protein expression level. In Fig 3b, we tried to investigate the effect of Flag-WDR77 on *ISRE* induction by individual molecule respectively, rather than compare the *ISRE* inductions by these molecules (MAVS, TBK1 or IRF3(S396D) etc.). In the revised version of our manuscript, we optimized the experimental conditions, including increased expression of IRF3(S396D) and decreased expression level of RIG-I(N), TBK1, and got a new set of data (**Supplementary Fig. 4b, c, Page 57**). The previous data were replaced with the new data.

Supplementary Fig. 4b, c

5. In Figure 4, TRAF3 was found associating with both WDR77 and MAVS, independently. What is the significance of TRAF3 in this association if they bind independently to both MAVS and WDR77? There does not seem to be any follow up on this in the subsequent parts, nor there is any studies on TRAF2. This seems puzzling.

Reply: We thank the reviewer for the critical comment. In Fig 3d, we examined the association of WDR77 with many signaling molecules and found that WDR77 co-immunoprecipitated only with MAVS and TRAF3 (Fig. 3d, Page 40). Therefore, we investigated the association between WDR77 and TRAF3 further in the context of virus infection. We found that the interaction between TRAF3 and WDR77 is independent of virus stimulation (Supplementary Fig. 5a, b, Page 59), in contrast to that the interaction between MAVS and WDR77 is significantly enhanced upon virus infection (Fig. 4a-c, Page 42). We further found that upon RNA virus stimulation, the interaction between TRAF3 and MAVS occurred independent of WDR77 (Fig. 4b, Page 42). Therefore, the interaction between WDR77 and TRAF3 may not be involved in the inhibition of WDR77 on MAVS aggregation. Since TRAF3 is a multi-functional molecule, its association with WDR77 may serve other cellular function and is beyond the scope of this study.

In addition, TRAF2 is a well-known binding partner of MAVS, and was used as a positive control in Figure 1 (Fig. 1e, Page 35).

6. The section title in line 260 reads, "Wdr77 is essential for antiviral immune response in mouse primary cells". How is Wdr77

essential for antiviral response when it is antagonist to the response itself?

Reply: We thank the reviewer for the comment and apologize for the inaccurate descriptions. They have been rephrased as: WDR77 deficiency enhances antiviral immune response in mouse primary cells (**Line 292, Page15**).

7. Line 272 reads “PEMs were prepared from WT, Wdr77CKO and Mavs^{-/-} mice followed by various treatments as indicated (Fig. 6a).” This statement seems to be wrong as the legend suggests that infections were performed in isolated cells, and not in animal after which the cells were isolated.

Reply: We thank the reviewer for the comment and apologize for the confusing descriptions in the original manuscript. They have been rephrased as: We prepared PEMs from WT, *Wdr77*^{CKO} and *Mavs*^{-/-} mice respectively. PEMs were then subjected to various treatments as indicated (Fig. 6a) (**Line 305, Page16**).

8. Though WDR77 was demonstrated as an antagonist to MAVS, how critical is WDR77 binding in inactivating MAVS during late stages of viral infection? There is no experimental set up that differentiates the dynamics of MAVS activation and inactivation during the whole process of infection, particularly between the early and the late stages of infection. It is merely proposed that WDR77 could be involved in the inactivation of MAVS signalling. However, more details on its temporal engagement with MAVS are required in order to consider it as an important player in muting MAVS signalling

Reply: We thank the reviewer for the critical comment. Following the reviewer's suggestions, we examined the dynamics of MAVS activation over a time course of 36 hours post virus infection with or without WDR77. We observed that both MAVS activation (as manifested by its aggregation) and IFNB induction (measured by qPCR) reached a peak at 16 hours, and decreased afterwards (**Fig. 4h, i, Page 42; Supplementary Fig. 5g, Page 59**). In the absence of WDR77, decrease of MAVS aggregation was much slower, indicating the downregulation of MAVS aggregation and activation by WDR77 in the later stage (i.e., from 16 hours in our specific experimental settings).

Meanwhile, the interaction between WDR77 and MAVS reached a peak at about 16 hours post virus infection (**Fig. 4c, Page 42**), correlated with the peak of MAVS aggregation.

9. What is the proposed mechanism by which WDR77 mutes MAVS signalling? Does WDR77 cause any PTM? how is its mode of action? Do WDR77 levels undergo dynamic changes during infection in order to inactivate MAVS?

Reply: We thank the reviewer for the critical comment. Our data showed that WDR77 binds to MAVS directly, through their WD2-WD3-WD4 and proline-rich region respectively (**Fig. 3f, g, Page 40**). In addition, we purified recombinant WDR77 and MAVS and showed that purified WDR77 can prevent the aggregation of MAVS particles and promote the disassembly of the prion-like

filament of MAVS *in vitro* (**Fig. 5c, d, Page 45**). These data suggested that WDR77 might mute MAVS signaling by binding to MAVS directly and preventing its aggregation. We propose a working model in the revised manuscript (**Supplementary Fig. 8g, Page 65**).

As an essential subunit of the 20S arginine methyltransferase methylosome complex, WDR77 might be involved in arginine methylation, since methylosome complexes are capable of catalyzing monomethylation and symmetric dimethylation of arginine¹. Previous reports have revealed that PRMT5 and WDR77 methylosomes cannot methylate MAVS proteins². On the other hand, though ubiquitination of MAVS might be involved in its aggregation, we did not detect any effect of WDR77 on MAVS in this regard. As we mentioned above, WDR77 can disassemble the prion-like filament formed by purified MAVS *in vitro* (**Fig. 5c, d, Page 45**). These data suggested that PTMs might not be involved in the inhibition of WDR77 on MAVS activation.

Our data showed that WDR77 level remains constant throughout infection (**Fig. 4a-c, g, h, Page 42; Supplementary Fig. 5b, f, Page 59**). In fact, more WDR77 is recruited to MAVS to play its inhibitory role post virus infection (**Fig. 3e, Page 40; Fig. 4a-c, Page 42**).

10. Though Wdr CKO confirmed that its loss indeed results in lower viral titers, it does not rule out the possibility that its effect is independent of MAVS. To further demonstrate that WDR77 imparts suppression of MAVS mediated antiviral signalling, these studies should be tested in WdrCKO + MAVS KO double KO.

Reply: We appreciate the valuable suggestions from the reviewer. Due to the time constraint, we were not able to get *Wdr77^{CKO}MAVS^{-/-}* double knockout mice. Hence, we utilized cell lines to address the reviewer's concern. We generated *WDR77^{-/-}MAVS^{-/-}* double knockout cells (**Supplementary 4d, Page 57**). Our data revealed that in *WDR77^{-/-}MAVS^{-/-}* cells, interferon induction upon SeV stimulation was completely abolished, suggesting that the increase of interferon induction in WDR77 knockout cells was indeed dependent on MAVS (**Supplementary 4e, f, Page 57**).

Minor points

1. Certain statements on WDR77 have been confusing at places. For eg., line 136 states that “To further validate its antiviral function, three WDR77 knockout HEK293T cell lines...”. While WDR77 is antagonistic to antiviral response and hence is proviral in function, it is referred to as “antiviral”. This needs to be addressed throughout the manuscript.

Reply: We thank the reviewer for the comment and apologize for the confusing descriptions in the original manuscript. They have been rephrased as: Furthermore, three WDR77 knockout cell lines (namely WDR77^{-/-} #1, #2, #3) were generated by the CRISPR/Cas9 technique (Line 139, Page 8). We have addressed similar issues throughout the manuscript.

2. Line 104-107: Figure 1C where WDR77 mw is 40kDa except that WDR77 molecular weight is close to 45kDa. Discuss the possible reason for this discrepancy.

Reply: We thank the reviewer for the comment. WDR77 is composed of 342 amino acids so that its molecular weight is presumably to be 44kDa. Many factors such as shape and PTMs might contribute to the apparent molecular weight manifested on SDS-PAGE. In our study, WDR77 appears to be around 42kDa when an anti-WDR77 antibody from Abcam (ab154190) was utilized. This discrepancy could be attributed to the inherent charge of WDR77, which may have an effect on its charge mass ratio but not on protein standard. The major band corresponding to 40kDa and subjected to mass spectrometry in Figure 1c (Fig. 1c, Page 35) was a mixture of many proteins with molecular weights about 40kDa, which contained WDR77.

3. Line 208 states “Based on these data, we speculate that WDR77

might be involved in the downregulation of MAVS activity after an effective innate immune response is launched to avoid harmful inflammation”. However, there is no immunoblot or any other data to show the status of viral infection and at what time WRD77 shows its maximum activity.

Reply: We appreciate the reviewer’s critical comment. As mentioned above, we examined the dynamics of MAVS activation over a time course of 36 hours post virus infection with or without WDR77. We observed that both MAVS activation and IFNB induction reached a peak at 16 hours, and decreased afterwards (**Fig. 4h-i, Page 42; Supplementary Fig. 5g, Page 59**). In the absence of WDR77, decrease of MAVS aggregation was much slower, indicating the downregulation of MAVS activation by WDR77 in the later stage (i.e., from 16 hours in our specific experimental settings). Meanwhile, the interaction between WDR77 and MAVS reached a peak at about 16 hours post infection (**Fig. 4c, Page 42**), correlated with the peak of MAVS aggregation.

4. Line 307 states “These data indicate that WDR77 is an important negative regulator of the antiviral immune response against RNA virus in vivo”. The authors have worked solely with VSV and SeV which are negative sense RNA viruses. Similar studies performed on a positive sense RNA virus can add further value to this study.

Reply: We thank the reviewer for the critical comment. Due to time constraints and that positive sense RNA virus is not available immediately in the lab, we cannot perform similar studies on any positive sense virus. Instead, we emphasize in the manuscript that our conclusions were obtained when a negative sense RNA virus was utilized (**Line 342, Page18; Line 369, Page19; Line 378, Page19**).

5. Line 334 states “Moreover, WDR77 deficiency resulted in enhanced activation of TBK1/IRF3 and upregulation of IFN- β induction in HEK293T, MEF cells and HeLa cells upon RNA virus but not DNA virus infection”. Since this was not tested using a DNA virus but only with HT-DNA as a stimulant, the authors need to change this statement.

Reply: We thank the reviewer for the critical comment and apologize for the inaccurate descriptions in the original manuscript. They have been rephrased

as: Moreover, WDR77 deficiency resulted in enhanced activation of TBK1/IRF3 and upregulation of IFN- β induction in HEK293T, MEF cells and HeLa cells upon negative sense RNA virus but not DNA stimulation (**Line 369, Page19**).

6. Line 710-713 : What is the duration of VSV infection in Figure 2g? The figure displaye12h while the legend shows 8h.

Reply: We thank the reviewer for the critical comment and apologize for the confusing descriptions in the original manuscript. The duration of VSV infection is 12h and we have corrected the mistake. We repeat the experiment to also include *MAVS*^{-/-} cells in the revised manuscript and detailed information is provided (**Line 747-752, Page 39; Fig. 2g-h, Page 38**).

7. Line 713-714; Figure 2h-j. Were these performed in 293 as mentioned in the figure or 293T cells? There is no mention about HEK293 in material method. KO cell lines generation also mention HEK293T, HeLa and MEF only. If it is HEK293 then it has to be mentioned in cells and KO generation section in material and methods.

Reply: We thank the reviewer for the critical comment and apologize for the missing information in the original manuscript. The experiment related to Figure 2i-k (**Fig. 2i-k, Page 38**) was indeed performed in HEK293 cells. We have provided detailed information for HEK293 in the materials and methods.

Reviewer #2

This manuscript by Li et al describes a negative regulation mechanism of MAVS signaling by WDR77. Through affinity pull-down the authors identified WDR77 as a protein that interacted with MAVS and describe negative regulation of MAVS-mediated antiviral signaling through the inhibition of MAVS aggregation. The authors followed up these in vitro results using a myeloid-specific WDR77 knockout mice to show enhanced resistance to VSV and IAV, but not HSV. The results convincingly show that WDR77 is indeed a negative regulator of the RIG-I-MAVS signaling, but the molecular mechanism that the authors propose are not always convincing and confusing. They claim that WDR77 is recruited to MAVS following activation of RLR signaling (pIC or

SeV treatment) Fig. 3 and 4. What is not clear from the results, how or what causes the stimulation of WDR77. Again, this model is inconsistent with results from Fig. 1 where expression and purification of HA-MAVS from unstimulated cells led to its identification as a MAVS interacting protein. Following are some suggestions for further experiments and major revision of this manuscript.

Reply: We thank the reviewer for the critical comment. We initially identified WDR77 as a MAVS-associated protein while purification of Flag-MAVS was performed in the absence of stimulation, suggesting that WDR77 might bind to MAVS without stimulation. In the following study, our data showed that WDR77 can indeed bind to MAVS in the absence of virus infection (zero time point), which is consistent with the purification result (**Fig. 3e, Page 40**). Moreover, we found that more WDR77 was recruited to MAVS post virus infection, which reached a peak at 16 hours and is correlated with MAVS aggregation (**Fig. 3e, Page 40; Fig. 4a-c, Page 42**). Following virus infection, the dynamic of MAVS aggregation is correlated with its binding to WDR77, suggesting that WDR77 binds to preferentially or with higher affinity to MAVS prion-like filament. Further structural study may provide more mechanistic insight into how WDR77 binds to MAVS filament and plays its inhibitory effect on IFNB induction post infection.

Major points:

1. Major editing of the manuscript is needed to make it clear and understandable.

Reply: We thank the reviewer for the critical reading of our manuscript. We have edited and polished the languish of the manuscript.

2. The manuscript focuses extensively on detecting MAVS aggregation and how it is inhibited by WDR77. However, some of these assays are highly qualitative such as AGE, and EM studies that are difficult to interpret without further supporting evidence. For example, in the EM studies it is not clear how are MAVS in peak 1 and peak 2 are dis-aggregating or aggregating spontaneously.

Reply: We thank the reviewer for the critical comment. Previous reports showed that purified recombinant MAVS contains two populations, which correspond to two peaks separated by sizing-exclusion chromatography^{3, 4}.

With the help of EM, it was found that Peak-1 contains prion-like filament and Peak-2 contains particle. Furthermore, a portion of MAVS in Peak-2 can spontaneously form prion-like filament after incubation for some time, while a portion of MAVS in Peak-1 can disassemble into shorter filament after incubation for some time (**Fig. 5c, d, Page 45**). These data suggested the two populations of recombinant MAVS are in a dynamic equilibrium. Our data showed that WDR77 can prevent the filament formation of MAVS in Peak-2 and promote the disassembly of MAVS filament in Peak-1.

3. Since RIG-I or MDA5 are the only known RNA sensors in this context, they might be involved in upstream sensing and initiating WDR77 recruitment to MAVS. This should be investigated. Alternatively, if WDR77 is pre-associated with MAVS, and help resolve the MAVS aggregates upon stimulation, it should be clearly demonstrated and explained.

Reply: We appreciate the valuable suggestions from this reviewer. We performed Co-IP experiment to examine the interaction between RIG-I, MDA5 and WDR77. Our data showed that WDR77 interacted with MAVS but not RIG-I or MDA5 (**Fig. 3d, Page 40; Fig. 4c, Page 42**), suggesting RIG-I or MDA5 may not play a role in WDR77 recruitment to MAVS.

On the other hand, more WDR77 is recruited to MAVS upon virus infection, suggesting that WDR77 binds to MAVS aggregates with higher affinity, which is consistent with the inhibition of WDR77 on MAVS aggregation following viral

stimulation (Fig. 3e, Page 40; Fig. 4a-c, Page 42).

4. Another relevant question in this context will be if WDR77 is a downstream target of MAVS signaling. In this scenario, it is conceivable that during the late stage of MAVS signaling WDR77 is transcribed, accumulated, and resolves MAVS aggregates to terminate the activation of RLR signaling.

Reply: We thank the reviewer for the critical comment. We notice that the protein level of WDR77 remains constant throughout the entire infection period as we examined (Fig. 4a-c, g, h, Page 42; Supplementary Fig. 5b, f, Page 59). In our model, WDR77 is recruited to MAVS aggregates by direct protein-protein interaction and negatively regulates its prion-like aggregation.

Reviewer #3

The manuscript by Li and colleagues from the Shanghai Institute of Molecular & Cellular Biology identified WDR77 as a MAVS-associating protein following a genetic screen using extract from mitochondria. WDR77 was shown to bind to the MAVS proline-rich region through its WD2-WD3-WD4 domain, and decreases RIG-I dependent signaling to the antiviral response. The data illustrates that WDR77 is recruited to MAVS to disassemble the formation of prion-like aggregates of MAVS. WDR77 deficiency generated by CRISP/R -cas results in a phenotype that potentiates induction of antiviral genes after VSV infection. Myeloid specific WDR77-deficient mice are more resistant to RNA virus infection.

It is difficult to identify a set of experiments that have not been completed in this study; many approaches have been taken – co-transfections to measure IFN response, challenge with VSV in the presence or absence of WDR77, knock-down of WDR77 or knock-out by CRISP/R-cas targeting are all included. The results argue that WDR77 is involved in termination of the antiviral response by dissociating the prion-like MAVS aggregates. The experiments are generally well performed and the English grammar and style are good – another round of proof-reading will improve the text. Several points for clarification:

Major comments:

1. Authors find only two proteins interacting with MAVS – WDR77 and PRMT5; could the authors discuss the reasons for this surprising finding – only 2 proteins interacting with MAVS. In fact many associated proteins have been described.

Reply: We appreciate the reviewer's comments. Our screening identified multiple proteins that may interact with MAVS, including some proteins that have been reported in previous literatures, whereas we did not provide a complete list of proteins identified. WDR77 turned out to be the most interesting one to us that plays an important role in regulating MAVS activity.

2. The authors imply a temporal and perhaps stochastic model of regulation - that is WDR77 appears/binds to MAVS later in infection to dissociate MAVS. Any evidence that WDR77 is induced at late times ? and in vivo is the amount of WDR sufficient to mediate these effects ?

Reply: We are grateful for the reviewer's insightful comment. Following the reviewer's suggestions, we examined WDR77 and MAVS over a time course of 36 hours post virus infection. WDR77 level was constant throughout the period of time, while MAVS activation reached a peak at 16 hours and decreased afterwards (**Fig. 4h-i, Page 42; Supplementary Fig. 5g, Page 59**). We further found that the interaction between WDR77 and MAVS reached a peak at about 16 hours post infection (**Fig. 4c, Page 42**), correlated with the peak of MAVS aggregation. Our data suggested that the amount of WDR77 is sufficient to mediate its inhibition on MAVS aggregation.

3. Throughout the manuscript, authors refer to the decrease in antiviral response as 'significantly increased' or 'dramatically decreased' but in fact the effects in down modulating antiviral activity is rather modest ~3 fold throughout. Is such a modest effect sufficient to block the complete antiviral response, or is WDR mechanism one of several interactions that contribute to MAVS dissociation?

Reply: We thank the reviewer for the critical comment. The effect on antiviral response observed could be determined by many factors, such as knockdown efficiency or expression level of WDR77, stimulants (virus stain or dsRNA mimics) and other experimental settings (treatment time, cell type etc.). We believe the effects we observed are all statistically significant. For example, as shown in Fig. 2g, VSV infectious titer from *MAVS*^{-/-} cells was about four folds of

that from WT cells, while VSV titer from WT cells was approximately nine folds of that from *WDR77*^{-/-} cells (**Fig. 2g, h, Page 38**), suggesting that *WDR77*^{-/-} cells are highly resistant to VSV proliferation. These data indicate that WDR77 is an important negative regulatory factor in inhibiting the RIG-I-MAVS signaling pathway in cells. Nevertheless, we cannot rule out the possibility that WDR77 mechanism could be one of several interactions that contribute to MAVS dissociation.

4. Human WD repeat domain 77 (*WDR77*, also *MEP50*) protein is a subunit of the 20S methylosome complex that includes type II protein arginine methyltransferase 5 (*PRMT5*), an enzyme responsible for mono- and symmetric dimethylation of arginine . The data does not clarify between a model in which WDR binds to MAVS vs a mechanism that is still dependent on *PRMT5* activity, but in association with *WDR77*. A knockdown of *PRMT5*, or co-expression of both molecules, or point mutations of *PRMT5* should help to clarify this point. Authors have likely performed such experiments. Essentially the authors have previously argued that MAVS regulation is dependent on *PRMT7* activity and now argue that regulation is dependent on *WDR77* without involvement of *PRMT5*.

Reply: We thank the reviewer for this critical comment and agree with the reviewer. Previous reports have demonstrated that the WDR77 is a subunit of the 20S methyl complex, which also includes PRMT5, an enzyme responsible for arginine monomethylation and symmetric dimethylation^{1, 5}. Indeed, we have performed experiments overexpressing both proteins. During our screening process, we simultaneously identified these two proteins. Overexpression of WDR77 but not PRMT5 suppressed IFNB induction, and overexpression of both WDR77 and PRMT5 did not generate a synergistic effect (**Fig. 1f-k, Page 35; Supplementary Fig. 1, Page 51**), suggesting that WDR77 but not PRMT5 is involved in RIG-I-MAVS-mediated IFNB induction. In the loss-of-function experiment, knockdown of either WDR77 or PRMT5 led to an increase of IFNB induction, which may be due to that endogenous WDR77 and PRMT5 are mutually dependent on each other (**Supplementary Fig. 2, Page 53; Fig. 2a, b, i, j, Page 38; Supplementary Fig. 3b, g, i, Page 55**). Furthermore, a previous report showed that PRMT5 and WDR77 methylosomes cannot methylate MAVS proteins². Together with previous reports, our data establish that WDR77, rather than PRMT5, inhibits the innate immune signaling pathway upon RNA virus infection. However, WDR77 may

not be the sole negative regulator of MAVS. In addition, a previous report showed that PRMT7 is also a negative regulatory factor⁶.

The observations of the manuscript are interesting but without a clarification of mechanism, the study is of limited significance, particularly since the authors previously argued for a different mechanism of PRMT action on MAVS.

Reply: We thank the reviewer for this critical comment. We would like to clarify here that we have not reported any mechanism on PRMT action on MAVS before. In this study, we found that WDR77 inhibits MAVS aggregation and disassemble its prion-like filament formation. It is a novel mechanism regulating MAVS activity upon virus infection. We respect previous reports on PRMT action on MAVS from other labs^{2, 6}.

References:

1. Antonysamy, S. et al. Crystal structure of the human PRMT5:MEP50 complex. Proc Natl Acad Sci U S A 109, 17960-17965 (2012).
2. Bai, X. et al. The protein arginine methyltransferase PRMT9 attenuates MAVS activation through arginine methylation. Nature Communications 13 (2022).
3. Hou, F. et al. MAVS forms functional prion-like aggregates to activate and propagate antiviral innate immune response. Cell 146, 448-461 (2011).
4. Qi, N. et al. Multiple truncated isoforms of MAVS prevent its spontaneous aggregation in antiviral innate immune signalling. Nat Commun 8, 15676 (2017).
5. Friesen, W.J. et al. A novel WD repeat protein component of the methylosome binds Sm proteins. J Biol Chem 277, 8243-8247 (2002).
6. Zhu, J. et al. Arginine monomethylation by PRMT7 controls MAVS-mediated antiviral innate immunity. Mol Cell 81, 3171-3186 e3178 (2021).

REVIEWERS' COMMENTS

Reviewer #2 (Remarks to the Author):

The primary question from all the reviewers in the first round of review related to the molecular mechanism of WDR77-mediated regulation of MAVS and its temporal nature. In my opinion, although the authors have addressed some of the critiques, the overall picture still remains unresolved without more detail studies.

Reviewer #3 (Remarks to the Author):

The authors have responded to the reviewers' requests with additional corrections to the text, new experiments and additional description of their results, all of which have improved the quality of the manuscript. The issue remains - is WDR77 interaction a significant mechanism of negative regulation of MAVS aggregation.

Reviewer #4 (Remarks to the Author):

The authors have appropriately addressed all the comments from the previous reviewers. The work presented in this manuscript is of high quality, novel and important. I support publication of this manuscript as is.

Response to reviewer's remarks

Reviewer #1

No comment to be addressed.

Reviewer #2

The primary question from all the reviewers in the first round of review related to the molecular mechanism of WDR77-mediated regulation of MAVS and its temporal nature. In my opinion, although the authors have addressed some of the critiques, the overall picture still remains unresolved without more detail studies.

Reply: We appreciate the reviewer's comments. Based on the data we have presented, we have concluded that WDR77 plays a crucial role as a regulator in the RIG-I-MAVS pathway by inhibiting MAVS aggregation. We plan to conduct further studies to explore its pathological relevance and gain more mechanistic insights through structural investigations. However, we acknowledge that these aspects are beyond the scope of this current study.

Reviewer #3

The authors have responded to the reviewers' requests with additional corrections to the text, new experiments and additional description of their results, all of which have improved the quality of the manuscript. The issue remains - is WDR77 interaction a significant mechanism of negative regulation of MAVS aggregation.

Reply: We appreciate the valuable comments provided by the reviewer. In Figure 3, we present compelling data illustrating the binding of WDR77 to MAVS through its WD2-WD3-WD4 domains. Notably, the overexpression of a WDR77 fragment (1-207) containing these domains leads to a significant inhibition of IFN-I induction. Furthermore, in Figure 5, we demonstrate that recombinant WDR77 can effectively inhibit the prion-like aggregation of MAVS *in vitro*, highlighting a mechanism facilitated by direct protein-protein

interaction. These findings strongly suggest that the interaction between WDR77 and MAVS plays a critical role in regulating MAVS aggregation.

Reviewer #4

The authors have appropriately addressed all the comments from the previous reviewers. The work presented in this manuscript is of high quality, novel and important. I support publication of this manuscript as is.

We thank this reviewer for the positive comment.